# Intrinsically disordered protein biosensor tracks the physical-chemical effects of osmotic stress on cells

Cesar L. Cuevas-Velazquez [1,2,3,4✉], Tamara Vellosillo[1,2], Karina Guadalupe[5,6], Hermann Broder Schmidt [7], Feng Yu [5,8], David Moses [5,6], Jennifer A. N. Brophy[1,2], Dante Cosio-Acosta[3], Alakananda Das[9], Lingxin Wang[9], Alexander M. Jones [10], Alejandra A. Covarrubias [3✉], Shahar Sukenik [5,6,8✉] & José R. Dinneny [1,2✉]

Cell homeostasis is perturbed when dramatic shifts in the external environment cause the physical-chemical properties inside the cell to change. Experimental approaches for dynamically monitoring these intracellular effects are currently lacking. Here, we leverage the environmental sensitivity and structural plasticity of intrinsically disordered protein regions (IDRs) to develop a FRET biosensor capable of monitoring rapid intracellular changes caused by osmotic stress. The biosensor, named SED1, utilizes the Arabidopsis intrinsically disordered AtLEA4-5 protein expressed in plants under water deficit. Computational modeling and in vitro studies reveal that SED1 is highly sensitive to macromolecular crowding. SED1 exhibits large and near-linear osmolarity-dependent changes in FRET inside living bacteria, yeast, plant, and human cells, demonstrating the broad utility of this tool for studying water-associated stress. This study demonstrates the remarkable ability of IDRs to sense the cellular environment across the tree of life and provides a blueprint for their use as environmentally-responsive molecular tools.

[1] Department of Biology, Stanford University, Stanford, CA 94305, USA. [2] Department of Plant Biology, Carnegie Institution for Science, Stanford, CA 94305, USA. [3] Departamento de Biología Molecular de Plantas, Instituto de Biotecnología, Universidad Nacional Autónoma de México, Cuernavaca, Morelos 62210, Mexico. [4] Departamento de Bioquímica, Facultad de Química, Universidad Nacional Autónoma de México, Ciudad de México 04510, Mexico. [5] Center for Cellular and Biomolecular Machines (CCBM), University of California, Merced, CA 95343, USA. [6] Chemistry and Chemical Biology Program, University of California, Merced, CA 95343, USA. [7] Department of Biochemistry, Stanford University School of Medicine, Stanford, CA 94305, USA. [8] Quantitative Systems Biology Program, University of California, Merced, CA 95343, USA. [9] Department of Molecular and Cellular Physiology, Stanford University, Stanford, CA 94305, USA. [10] Sainsbury Laboratory, Cambridge University, Cambridge CB2 1LR, UK. ✉email: cuevas@quimica.unam.mx; alejandra.covarrubias@ibt.unam.mx; ssukenik@ucmerced.edu; dinneny@stanford.edu

ntracellular osmotic fluctuations are one of the most common physical–chemical perturbations cells experience throughout their life. In the absence of external stressors, the metabolic activity of the cell can induce large changes in the concentration of different metabolites that alter intracellular osmolarity[1]. Additional osmotic variation can be caused by the activity of ion channels that change the total concentration of free inorganic ions ($K^+$, $Na^+$, $Mg^{2+}$, etc.)[2]. Severe intracellular osmotic perturbations are readily caused by environmentally induced stress conditions, where the osmolarity outside of the cells changes dramatically. For instance, a decrease in water content in the exterior of a cell increases extracellular osmolarity in a way that causes the passive efflux of water out of the cell. This results in an immediate collapse of cell volume and concomitant increase in the concentration of solutes, macromolecular crowding, and the viscosity of the cell interior, impacting various molecular and cellular functions[3–5].

Despite the importance of osmotic regulation on cellular function, our mechanistic understanding of how cells sense such conditions, particularly in multicellular organisms, is limited[6,7]. One of the main barriers to better understanding the intracellular effects of osmotic stress is the lack of methods to reliably monitor physical–chemical changes that occur in single cells, in real time, and in a non-destructive manner[6,8,9].

Genetically encoded fluorescent biosensors are optical tools that enable the dynamic visualization and quantification of biochemical events that occur in living cells at various scales, from single cells to whole organisms[10]. Fluorescent biosensors are chimeric proteins composed of at least one fluorescent protein fused to a sensing domain. The selection of the sensing domain is based on its ability to specifically change its conformation in the presence or absence of an analyte[11]. The conformational change of the sensing domain then causes a change in the fluorescence readout that can be quantified. As of today, there are dozens of different fluorescent biosensors used to track small molecules, phosphorylation events, neurotransmitters, posttranslational modifications, and hormones; however, just a small fraction of biosensors are designed to report changes in the physical–chemical properties of the environment[12–16]. The main challenge for developing environmentally responsive biosensors is in sourcing sensory domains capable of specifically and reversibly altering their structure in response to changes in the physical-chemical properties of the cell.

Intrinsically disordered regions (IDRs) are protein domains that lack a stable three-dimensional structure and instead behave as ensembles of dynamic and rapidly changing conformations[17]. Because IDRs have a more extended surface area than globular proteins, they are highly sensitive to the physical–chemical properties of the solvent. Conditions such as pH, temperature, redox state, and high osmolarity induce conformational changes in some IDRs[18]. Recent work shows that environmental sensitivity is a shared property of many IDRs[19–21]. Furthermore, it has been proposed that the environmental sensitivity of IDRs could be used to regulate their activity, potentially allowing them to function as sensors of the environment[9,22,23]. Based on the aforementioned properties, we propose that IDRs are promising candidates for designing environmentally responsive biosensors.

Here, we demonstrate the use of IDRs for the development and implementation of a Förster Resonance Energy Transfer (FRET) biosensor that tracks the effects of osmotic stress on living cells. The biosensor, named SENSOR EXPRESSING DISORDERED PROTEIN 1 (SED1), dynamically monitors the response of budding yeast to osmotic stress at the cellular level. SED1 can also be used to track the effects of osmotic stress on living bacteria, plant, and human cells. We anticipate that the use of IDR-based fluorescent biosensors such as SED1 will aid in understanding how cells sense, respond, and acclimate to dynamic

environmental fluctuations caused by water-associated stress and other conditions.

## Results

### Design of a biosensor for studying the effects of osmotic stress on living cells

To track the effects of osmotic stress on living cells, we sought to combine the power of osmo-sensitive IDRs and ratiometric FRET readouts to build a genetically encoded fluorescent biosensor. For the sensory domain, we tested two members of the group 4 LATE EMBRYOGENESIS ABUNDANT (LEA) proteins from the model plant *Arabidopsis thaliana*[24]. Group 4 LEA proteins are intrinsically disordered proteins that exhibit a reversible disorder-to-folded transition in response to increased osmolarity in vitro[25]. We hypothesized that such osmolarity-dependent conformational changes would also occur inside living cells, making them excellent candidates for environmentally responsive biosensor development.

To test the ability of group 4 LEA protein structure to change in response to osmotic stress in vivo, we fused either *AtLEA4-2* or *AtLEA4-5* ORFs between the coding sequences of a FRET-compatible pair of fluorophores (mCerulean3 as the donor and Citrine as the acceptor) (Fig. 1a). We expressed the constructs in budding yeast (*Saccharomyces cerevisiae*) and treated the cells with NaCl to induce hyperosmotic shock. Both constructs exhibited a significant NaCl-concentration-dependent increase in the acceptor-to-donor emission ratio (Fig. 1b). We observed that the treatment displayed typical FRET behavior with an increase in fluorescence intensity of the acceptor (donor excitation-acceptor emission; DxAm) coupled to a decrease in fluorescence intensity of the donor (donor excitation-donor emission; DxDm) (Fig. 1c and Supplementary Fig. 1a), leading to a higher acceptor to donor emission ratio (DxAm/DxDm) (Fig. 1b). The FRET ratio change was significantly smaller when we tested a globular protein (arabinose-binding protein, ABP) as a reference[26] (Fig. 1b and Supplementary Fig. 1b). We chose ABP because it is a globular protein that has been successfully used in a biosensor of a small molecule (arabinose) with a small $K_d$, and we expected it would be insensitive to high osmolarity and/or macromolecular crowding. Hyperosmotic treatment with increasing concentrations of other ionic (Supplementary Fig. 1c) and non-ionic (Supplementary Fig. 1d) osmolytes showed that the change in FRET of both constructs was osmolarity-dependent and not osmolyte-specific (Supplementary Fig. 1e, f). Since AtLEA4-5 exhibited the largest FRET change in response to osmotic shock, we continued our characterization with this construct. The fluorescence intensity of single mCerulean3 or Citrine fused to AtLEA4-5 was not significantly affected by hyperosmotic shock induced with different osmolytes, demonstrating the stable fluorescence emission of the fluorophores in such conditions (Supplementary Fig. 1g, h). Finally, testing different FRET pairs confirmed the large dynamic range of mCerulean3-Citrine pair (Supplementary Fig. 2).

We searched for the sequence determinants of AtLEA4-5 environmental responsiveness. To do so, we split the full sequence into its two functional domains (termed N-terminal conserved domain and C-terminal variable domain, based on their sequence similarity to other group 4 LEA proteins), individually introduced them into the FRET system, and expressed them in live yeast cells. The N-terminal conserved domain has the ability to fold into an α-helix upon increased osmolarity in vitro, while the C-terminal variable domain remains largely disordered[25], suggesting that deletion of the latter may enhance the osmo-sensitivity of our reporter. However, we found that full-length AtLEA4-5 is necessary to reach the highest FRET ratio change upon hyperosmotic treatment (Fig. 1d), suggesting

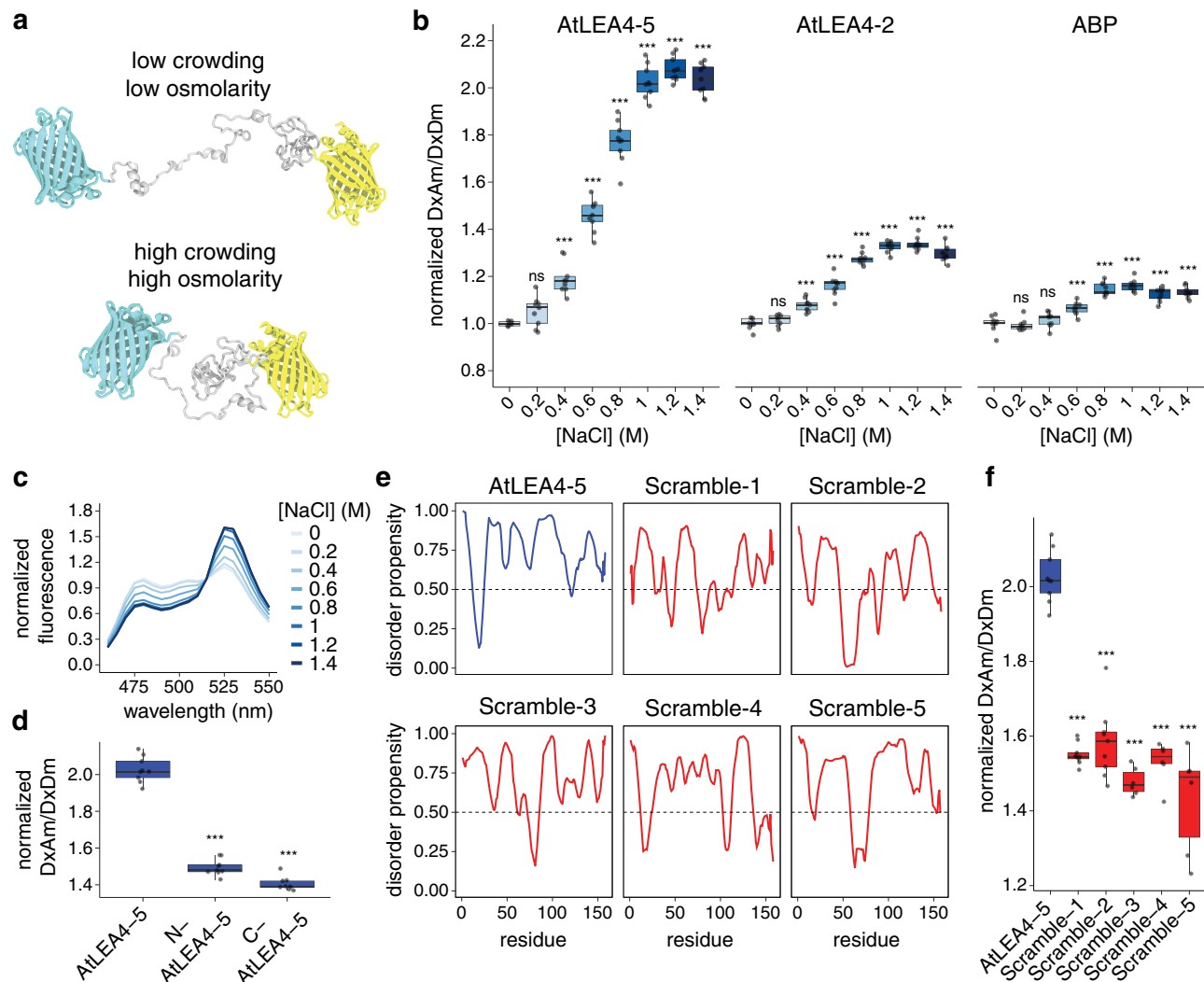

**Fig. 1 Design of a biosensor for studying the effects of osmotic stress on living cells. a** Schematic representation of the biosensor design under low and high macromolecular crowding/osmolarity—prevalent intracellular conditions upon hypoosmotic or hyperosmotic stress, respectively. The conformations are selected from the ensemble of all-atom simulations of AtLEA4-5 in the corresponding conditions. Cyan: mCerulean3. Yellow: Citrine. Gray: AtLEA4-5. **b** Normalized FRET ratio (DxAm/DxDm) of live yeast cells treated with different concentrations of NaCl. Cells are expressing the biosensor construct using either AtLEA4-5, AtLEA4-2, or arabinose-binding protein (ABP) as the sensory domain. $n = 9$ independent measurements. Two-way ANOVA. $*p < 0.05$, $**p < 0.01$, $***p < 0.001$. Boxes represent 25th–75th percentile (line at median) with whiskers at 1.5*IQR. **c** Fluorescence emission spectra of NaCl-treated live yeast cells expressing the biosensor construct using AtLEA4-5 as the sensory domain. Fluorescence values were normalized to the value at 515 nm. **d** Normalized FRET ratio (DxAm/DxDm) of live yeast cells expressing either AtLEA4-5, N-AtLEA4-5, or C-AtLEA4-5 biosensor constructs. Cells were treated with 1 M NaCl. $n = 9$ independent measurements. One-way ANOVA. $*p < 0.05$, $**p < 0.01$, $***p < 0.001$. Boxes represent 25th–75th percentile (line at median) with whiskers at 1.5*IQR. **e** Disorder propensity prediction of AtLEA4-5 (blue) and five different scrambled versions (red) using PONDR. Threshold at 0.5 disorder propensity is shown. **f** Normalized FRET ratio (DxAm/DxDm) of live yeast cells expressing AtLEA4-5 (blue) or five different scrambled versions (red). $n = 9$ independent measurements. One-way ANOVA. $*p < 0.05$, $**p < 0.01$, $***p < 0.001$. Boxes represent 25th–75th percentile (line at median) with whiskers at 1.5*IQR. Source data are provided as a Source Data file.

that both domains are required for the full conformational change in vivo.

Next, we tested how the primary amino acid sequence and amino acid composition of AtLEA4-5 affected the dynamic FRET properties of our reporter. We synthesized five different scrambled versions of the AtLEA4-5 coding sequence holding the amino acid composition and length constant (Supplementary Fig. 3a). We designed the scrambled versions to remain highly disordered, but with a decreased propensity, relative to AtLEA4-5, to form an α-helix (Fig. 1e and Supplementary Fig. 3b). Additionally, we selected the scrambled versions that displayed a smaller degree of charge mixing than AtLEA4-5, as denoted by larger Kappa values (Supplementary Fig. 3c)[27]. When these

sequences were used to generate FRET reporters and expressed in yeast, we found that the different scrambled versions displayed a diminished magnitude of FRET response to hyperosmotic stress compared to the native AtLEA4-5 sequence (Fig. 1f). Together these results suggest that the AtLEA4-5 protein is able to undergo osmotic-stress-induced conformational changes in vivo, and that these changes are dependent on the full-length primary amino acid sequence of the protein.

**AtLEA4-5 is highly sensitive to the chemical composition of the solution.** As water leaves the cell during hyperosmotic shock, a number of physical-chemical properties of the intracellular environment change; in particular, the concentration of organic

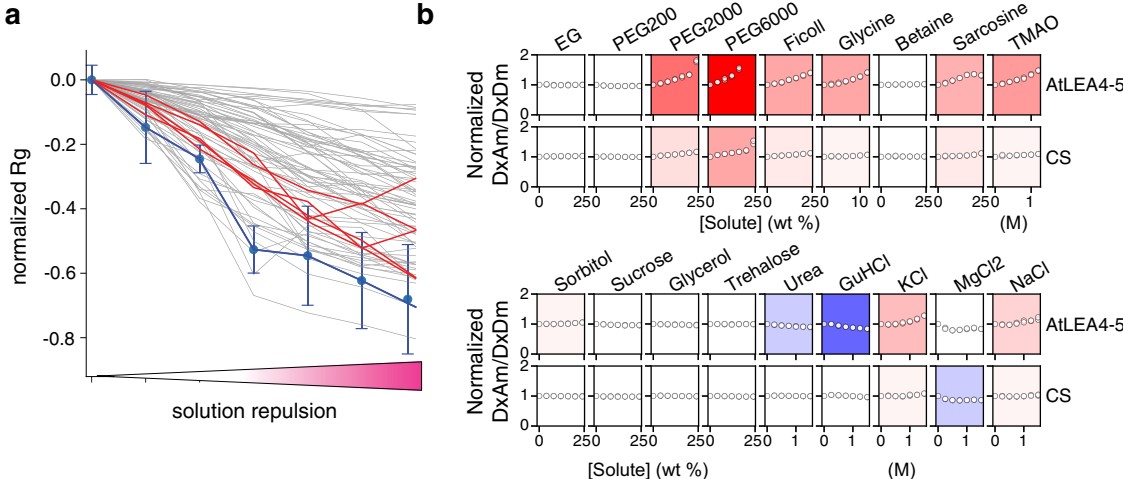

**Fig. 2 AtLEA4-5 is highly sensitive to the chemical composition of the solution. a** Computational solution space scan of the normalized radius of gyration (Rg) of AtLEA4-5 (blue), five different scrambled sequences shown in Fig. 1e, f (red), and 70 different naturally occurring IDRs (gray) under different solution repulsion levels (low to high solution repulsion of the protein backbone). Mean ± SD from $n = 5$ independent simulations. For clarity, only AtLEA4-5 (blue) SD is shown. SD for the other proteins are provided in the Source Data file. **b** Experimental solution space scan of AtLEA4-5 and CS. Open circles show the normalized FRET ratio (DxAm/DxDm) for the indicated concentration of each solute, with two points (that often overlap) for each concentration taken from separate repeats, highlighting the reproducibility of the data. Background color intensity represents sensitivity to the addition of solute. Stronger colors indicate stronger sensitivity. Red: compaction; blue: expansion; white: no change. Solution concentrations are given in weight percent (0–25 or 0–12 wt%) or molar (0–1.5 M). Source data are provided as a Source Data file.

and inorganic solutes rises, as does the extent of macromolecular crowding. Any of these properties could underlie the biophysical mechanism driving the conformational changes in AtLEA4-5. Macromolecular crowding is a general condition of the cell interior that gets exacerbated under hyperosmotic conditions due to water loss[28]. To further investigate the mechanism of AtLEA4-5 responsiveness observed in cells, we designed an approach to test AtLEA4-5 sensitivity to different solutions in silico and in vitro.

First, we performed all-atom Monte Carlo simulations to sample the conformational landscape of AtLEA4-5 under a wide range of solution conditions. This class of simulation, known as solution space scanning, has been used to investigate the solution-protein interactions of dozens of IDRs[19,20]. We used this method to exert a compacting force on a range of IDRs and compared the tendency of the different sequences to compact. We observed that AtLEA4-5 showed an enhanced sensitivity to such compaction compared to the scrambled versions of the sequence, in agreement with our in vivo observations (Figs. 2a, 1f). Furthermore, a comparison with 70 different naturally occurring IDRs[20] showed that AtLEA4-5 was an outlier in terms of its high sensitivity (Fig. 2a).

Next, we investigated the solution sensitivity of AtLEA4-5 in vitro. We used the FRET efficiency of recombinantly expressed and purified full-length AtLEA4-5 fused to mCerulean3 and Citrine as a proxy for the end-to-end distance of the construct under different solution conditions. We induced macromolecular crowding with solutions of different molecular weight polyethylene glycol (PEG) isoforms at various concentrations, and compared these results to a previously reported macromolecular crowding biosensor (CS) as a reference[14]. The CS sensory domain is a synthetic, helical peptide with a hinge-like topology thought to compact in response to higher macromolecular crowding. Our experiments showed that PEG induced the compaction of AtLEA4-5 in a concentration and size-dependent manner (Supplementary Fig. 4). The PEG-induced compaction was more prominent in AtLEA4-5 than in CS, confirming the relative sensitivity of AtLEA4-5 to macromolecular crowding. This

observation was confirmed with Ficoll, another type of macromolecular crowding agent (Fig. 2b). Further characterization with a diverse set of osmolytes and salts (experimental solution space scan) revealed that AtLEA4-5 is particularly sensitive to amine-based osmolytes such as glycine, sarcosine and TMAO, but not betaine (Fig. 2b). Finally, we found that ionic strength is unlikely to play a major role in AtLEA4-5 compaction since NaCl and KCl showed only a modest effect even at very high concentrations. Together, these data show that despite its intrinsic disorder, the conformational ensemble of AtLEA4-5 is highly responsive to changes in the chemical composition of the solution, particularly macromolecular crowding and amine-based osmolytes, in silico and in vitro, and that these properties are based on both topology and amino acid sequence.

**SED1 can dynamically track the effects of osmotic stress on live yeast cells.** We renamed the transgene expressing AtLEA4-5 to *SENSOR EXPRESSING DISORDERED PROTEIN 1* (*SED1*)—"sed" translates into "thirst" in Spanish. When live yeast cells expressing SED1 were treated with NaCl and followed over time, we found that the FRET response was fast and reversible, and allowed the acclimation of yeast cells to be measured over time after hyperosmotic shock (Fig. 3a). We then analyzed SED1 performance as an osmotic/crowding biosensor using CS as reference in vivo. We found that the dynamic range of SED1 was larger than that of CS (Fig. 3b). This was consistent with our observations in vitro (Fig. 2b), demonstrating the improved performance of IDR-based biosensors over existing tools[14].

To further validate the sensitivity of our biosensor in vivo, we sought to genetically interfere with a well-characterized osmo-responsive pathway in yeast. To this end, we expressed SED1 in *hog1Δ* and *pbs2Δ* yeast KO mutant backgrounds. These mutants disrupt key components of the HOG (High-Osmolarity Glycerol) pathway, which is activated in yeast to respond and acclimate to increased osmolarity of the surrounding medium[29]. Pbs2 is a scaffold MAPKK that integrates the two branches of the HOG pathway[30]. Hog1 is a MAPK that, upon hyperosmotic shock,

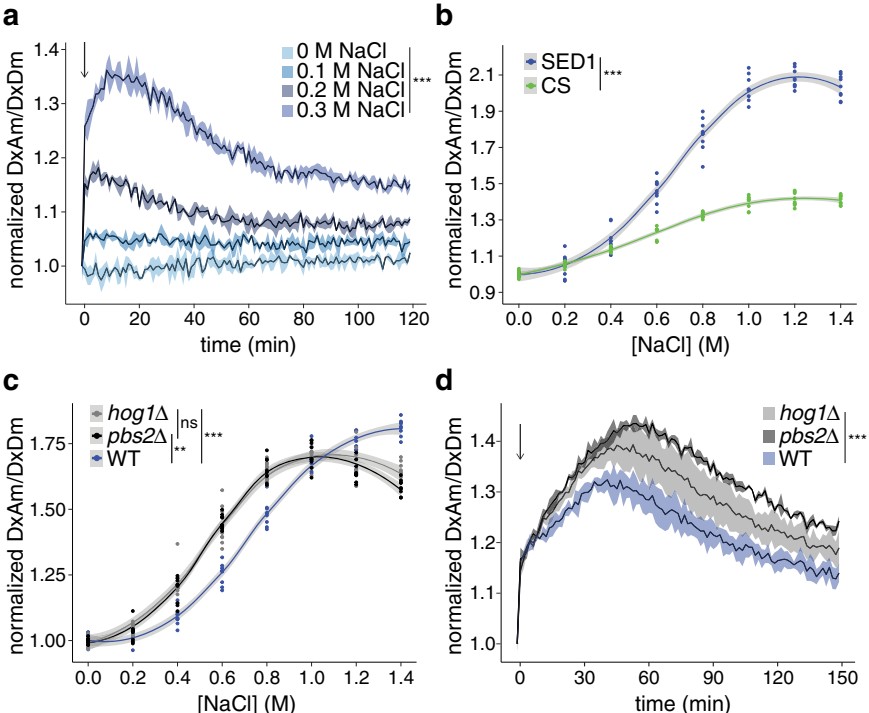

**Fig. 3 SED1 can dynamically track the effects of osmotic stress on live yeast cells. a** Normalized FRET ratio (DxAm/DxDm) time course of live yeast cells expressing SED1, treated with different concentrations of NaCl. The arrow indicates the addition of the treatment. Mean ± SEM. One-way ANOVA. ***$p < 2 \times 10^{-16}$. **b** Normalized FRET ratio (DxAm/DxDm) of live yeast cells expressing SED1 (blue) and CS (green), treated with different concentrations of NaCl. One-way ANOVA. ***$p < 2 \times 10^{-16}$. Continuous lines were smoothed using R with a loess smoothing function. Shaded regions indicate 95% confidence intervals. **c** Normalized FRET ratio (DxAm/DxDm) of wild type BY4742 strain (blue), hog1Δ::G418 mutant (gray), and pbs2Δ::G418 mutant (black), live yeast cells expressing SED1, hyperosmotically shocked with different concentrations of NaCl. Measurements were done immediately after hyperosmotic shock. Two-way ANOVA. *$p < 0.05$, **$p < 0.01$, ***$p < 0.001$. Continuous lines were smoothed using R with a loess smoothing function. Shaded regions indicate 95% confidence intervals. **d** Normalized FRET ratio (DxAm/DxDm) time course of wild type BY4742 strain (blue), hog1Δ::G418 mutant (gray), and pbs2Δ::G418 mutant (black), live yeast cells expressing SED1, treated with 0.6 M sorbitol. The arrow indicates the addition of the treatment. Mean ± SEM. Two-way ANOVA. ***$p < 2 \times 10^{-16}$. Source data are provided as a Source Data file.

translocates to the nucleus to activate transcription factors that induce genes required to promote osmoprotectant glycerol accumulation and osmotic acclimation[31]. Yeast mutants of these genes are sensitive to hyperosmotic stress and cannot appropriately acclimate to these conditions[32]. When measured a few seconds after treatment with concentrations lower than 1 M NaCl, SED1 FRET ratio was larger in hog1Δ and pbs2Δ mutants than in the wild type (WT) (Fig. 3c). The opposite occurred when the NaCl concentration was higher than 1 M. Since wild type cells respond and acclimate faster than the mutants under mild hyperosmotic shock (<1 M NaCl), our data suggest that SED1 response reflects the decrease in intracellular osmolarity/crowding resulting from the acclimation process.

Next, we followed the SED1 FRET ratio over time in the different genetic backgrounds after hyperosmotic shock with 0.6 M sorbitol. As expected, we observed an immediate increase in SED1 FRET ratio upon hyperosmotic shock in all the genotypes; however, in contrast to the wild type, the mutants displayed a sustained increase in FRET before declining (Fig. 3d), consistent with the reduced ability of these genotypes to acclimate. These data underscore the sensitivity of SED1 to osmotic stress in cells and suggests that it can be used to characterize the physiological effects of genetic mutants disrupting well-studied and novel osmotic stress response pathways.

**Tracking SED1 response to osmotic stress in single cells reveals that vacuoles buffer against water loss.** Single-cell measurements allow researchers to resolve the heterogeneity that arises in cell

populations. The power of single-cell genomics has revealed the cell-type-specificity of a variety of molecular and physiological responses[33]. Molecular tools that allow single-cell resolution measurements will pave the way for unraveling currently overlooked biological mechanisms. Fluorescence biosensors are intrinsically suitable for investigating biological processes with single-cell resolution using microscopy, so we aimed to investigate the performance of SED1 in individual cells. We observed that the FRET ratio in individual cells increased when they were treated with 0.5 M NaCl, in agreement with our population measurements (Fig. 4a, b and Fig. 1b). Interestingly, the FRET ratio varied between cells, even under non-stress conditions (Fig. 4a, b), and correlated with sensor expression (Supplementary Fig. 5a, b). These data suggested that sensor expression, and overall protein concentration, may correlate with macromolecular crowding in the cell. In support of this hypothesis, experiments using purified SED1 found no correlation between SED1 concentration and FRET ratio in vitro (Supplementary Fig. 5c).

We also observed that SED1 protein localization was altered by hyperosmotic stress treatment. SED1 was homogeneously distributed throughout the cytoplasm under standard conditions, and rapidly re-distributed into spherical-shaped granules under 0.5 M NaCl (Supplementary Fig. 5d). Such granules were only observed in yeast cells. We found that the hyperosmotic-induced FRET ratio increase was not caused by the formation of SED1 granules, since treatment with 1,6-hexanediol, a compound often used to dissociate liquid-like condensates[34], dissociated the SED1 granules but did not decrease the FRET response (Supplementary Fig. 5e, f).

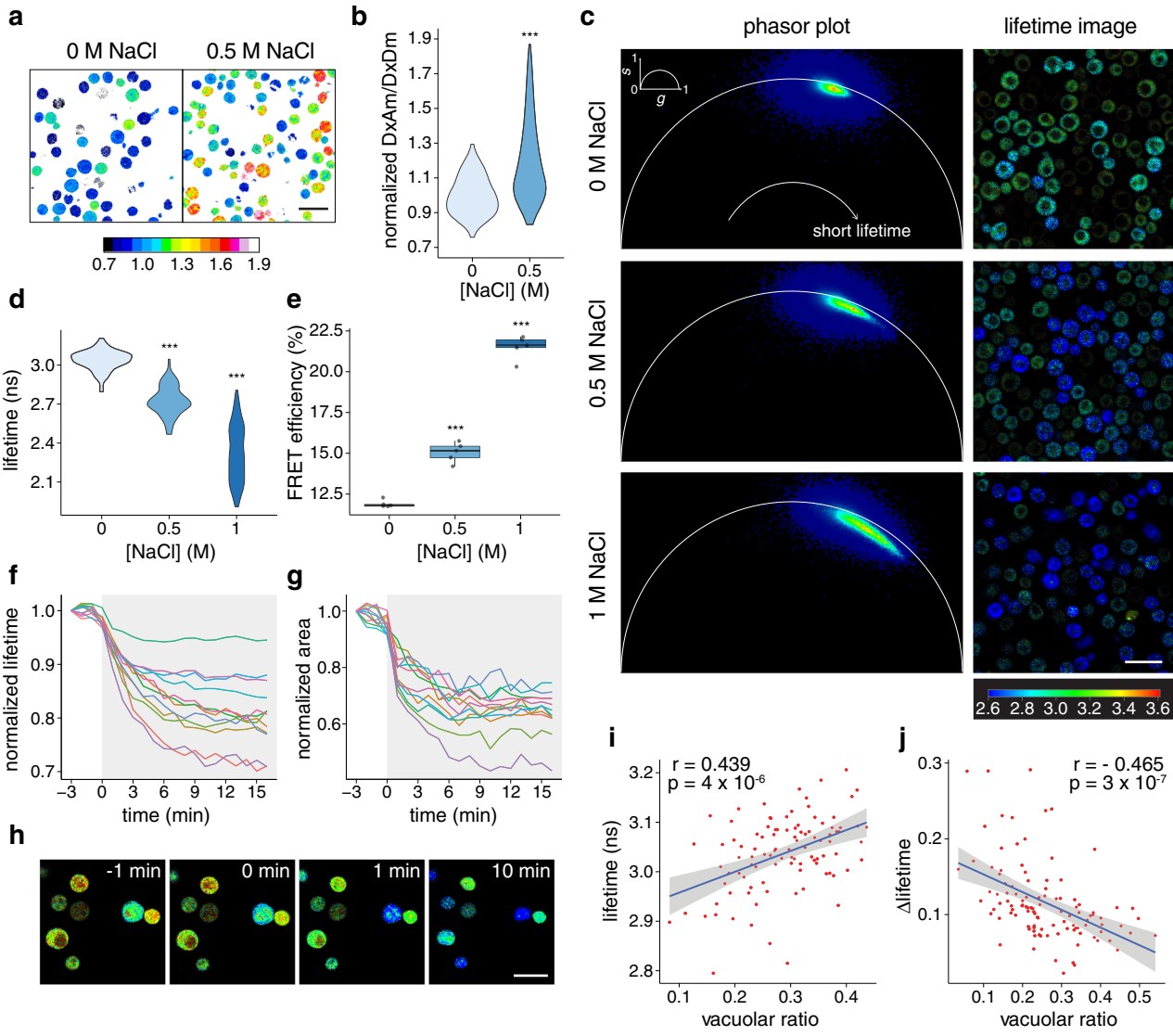

**Fig. 4 Tracking SED1 response to osmotic stress in single cells reveals vacuoles buffer against water loss. a** Ratiometric image of live yeast cells expressing SED1 under 0 M and 0.5 M NaCl. Scale bar = 10 μm. Calibration bar represents the normalized FRET ratio (DxAm/DxDm). **b** Quantification of (**a**). $n = 40$ cells (0 M NaCl) and $n = 67$ cells (0.5 M NaCl). Two-sided Student's $t$ test. ***$p = 1 \times 10^{-15}$. **c** Phasor plots (left) and donor fluorescence lifetime images (right) of live yeast cells expressing SED1 under 0 M, 0.5 M, and 1 M NaCl. Signals shifted to the left side of the phasor plot represent longer fluorescence lifetimes, whereas signals shifted to the right side represent shorter fluorescence lifetimes. Scale bar = 10 μm. Calibration bar represents the donor fluorescence lifetime in nanoseconds (ns). **d** Quantification of the donor fluorescence lifetime of individual cells from images in (**c**). $n = 100$ cells per treatment. One-way ANOVA. ***$p < 2 \times 10^{-16}$. **e** FRET efficiencies of live yeast cells from images in (**c**). $n = 5$ images for each treatment. One-way ANOVA. ***$p < 1 \times 10^{-11}$. Boxes represent 25th–75th percentile (line at median) with whiskers at 1.5*IQR. **f** Normalized donor fluorescence lifetime measured for single cells after 1 M NaCl treatment (shaded area) in a time course. The experiment was repeated 3 times with similar results. **g** Normalized area measured for single cells after 1 M NaCl treatment (shaded area) in a time course (same cells as in (**f**)). The same color represents the same cell for (**f**) and (**g**). The experiment was repeated 3 times with similar results. (**h**) Individual time frames showing the donor fluorescence lifetime of single yeast cells exposed to 1 M NaCl treatment at time 0 min. Scale bar = 10 μm. The calibration bar is the same as in (**c**). The experiment was repeated 3 times with similar results. **i** Pearson's correlation of donor lifetime and vacuolar ratio values for single yeast cells under standard conditions (0 M NaCl). Pearson's correlation coefficient $r = 0.439$, $p$-value $= 4 \times 10^{-6}$. Continuous line was smoothed using R with a linear method smoothing function. Shaded region indicates 95% confidence interval. **j** Pearson's correlation of the change in donor lifetime (Δlifetime) and vacuolar ratio values for single yeast cells subjected to 1 M NaCl. Δlifetime = (final lifetime−initial lifetime)/initial lifetime. Pearson's correlation coefficient $r = -0.465$, $p$-value $= 3 \times 10^{-7}$. Continuous line was smoothed using R with a linear method smoothing function. Shaded region indicates 95% confidence interval. Source data are provided as a Source Data file.

To further confirm that the response of SED1 to hyperosmotic stress was caused by *bona fide* donor-to-acceptor FRET, we performed fluorescence lifetime imaging-FRET (FLIM-FRET) experiments. FLIM is not sensitive to fluctuations in biosensor concentration, shading, excitation intensity, or background noise caused by the light source[35]. In FLIM-FRET experiments, the fluorescence lifetime of the donor is effectively decreased when it undergoes FRET with the acceptor[35]. AtLEA4-5 fused to mCerulean3 (donor-only control) expressed in yeast cells under non-stress conditions had an average fluorescence lifetime of $3.62 \pm 0.10$ nanoseconds (ns) (Supplementary Fig. 6). The average fluorescence lifetime of mCerulean3 in the SED1 construct under

non-stress conditions was $3.04 \pm 0.10$ ns, indicating a basal FRET efficiency of $12 \pm 1\%$ (Fig. 4c–e).

Hyperosmotic shocks with increasing concentrations of NaCl induced a progressive shift of the cell population to shorter fluorescence lifetimes, reaching an average of $2.32 \pm 0.21$ ns and an average FRET efficiency of $21 \pm 1\%$ under 1 M NaCl (Fig. 4c–e). Time-lapse imaging of individual SED1-expressing cells revealed that the drop in fluorescence lifetime occurs concomitantly with the reduction of cell volume caused by hyperosmotic shock (Fig. 4f–h and Supplementary Video 1), demonstrating the fine temporal resolution capabilities of SED1 and confirming its donor-to-acceptor FRET behavior.

We further investigated the cellular basis for FRET heterogeneity between cells (Fig. 4a–d). We noticed that cells with large visible vacuoles tend to have longer donor lifetimes under non-stress conditions (Fig. 4c). To confirm this, we quantified the vacuolar ratio per cell, that is, the proportion of the total cell area that is occupied by the vacuole (vacuolar ratio = vacuole area/cell area). We found that the vacuolar ratio had a significant positive correlation with the donor fluorescence lifetime even in the absence of stress (Fig. 4i). No correlation was observed between the cell area and donor lifetime, confirming the dominant effect of the vacuolar ratio over the FRET signal of SED1 (Supplementary Fig. 7a). We then measured the magnitude of the SED1 FRET change in individual cells upon hyperosmotic stress using time-lapse imaging. We found that neither the initial cell area nor the magnitude of cell area change predicted the fluorescence lifetime change upon stress (Supplementary Fig. 7b, c). Strikingly, however, we observed a significant negative correlation between the change in fluorescence lifetime after treatment and the vacuolar ratio before stress (Fig. 4j), suggesting that large vacuoles might effectively buffer water loss during hyperosmotic shock. Overall, the use of SED1 revealed how rapid cell-specific changes in the osmotic status of cells are well correlated to intracellular features such as relative vacuolar size. Furthermore, our data demonstrate the importance of measuring physical–chemical properties at the single-cell level for obtaining mechanistic insights of cellular homeostasis.

**SED1 tracks changes in osmolarity in a wide set of organisms**. Given the ability of SED1 to report the effects of osmotic stress on budding yeast, we sought to apply it to other biological systems. We first expressed SED1 in the bacteria *Escherichia coli*. Similar to what we found in budding yeast, we observed a hyperosmotic stress-dependent increase in the FRET readout (Fig. 5a and Supplementary Fig. 8a). Next, we tested SED1 in two evolutionarily distant multicellular organisms: plants and humans.

Plants heavily rely upon water to provide structural support and to facilitate gas exchange with the environment[9]. To test the utility of SED1 in this context, we transiently expressed a nuclear-localized SED1 transgene in tobacco leaves. Small discs of leaf tissue were placed onto 96-well plates, in wells containing hyperosmotic (sorbitol or NaCl) or hypoosmotic (water) solutions. We found that when SED1-expressing leaf discs were incubated with sorbitol or NaCl, the FRET readout increased over time (Fig. 5b), with an increase in fluorescence intensity of the acceptor and a concomitant decrease in fluorescence of the donor (Supplementary Fig. 8b, c). On the other hand, when leaf discs were treated with pure water, the opposite behavior in FRET and fluorescence intensities was observed (Fig. 5b and Supplementary Fig. 8d). These results indicate that SED1 is functional in multicellular photosynthetic organisms and encouraged us to further characterize SED1 in *Arabidopsis thaliana* transgenic lines. Arabidopsis seedlings expressing *pUBQ10::nlsSED1* were imaged before and after the addition of a solution containing

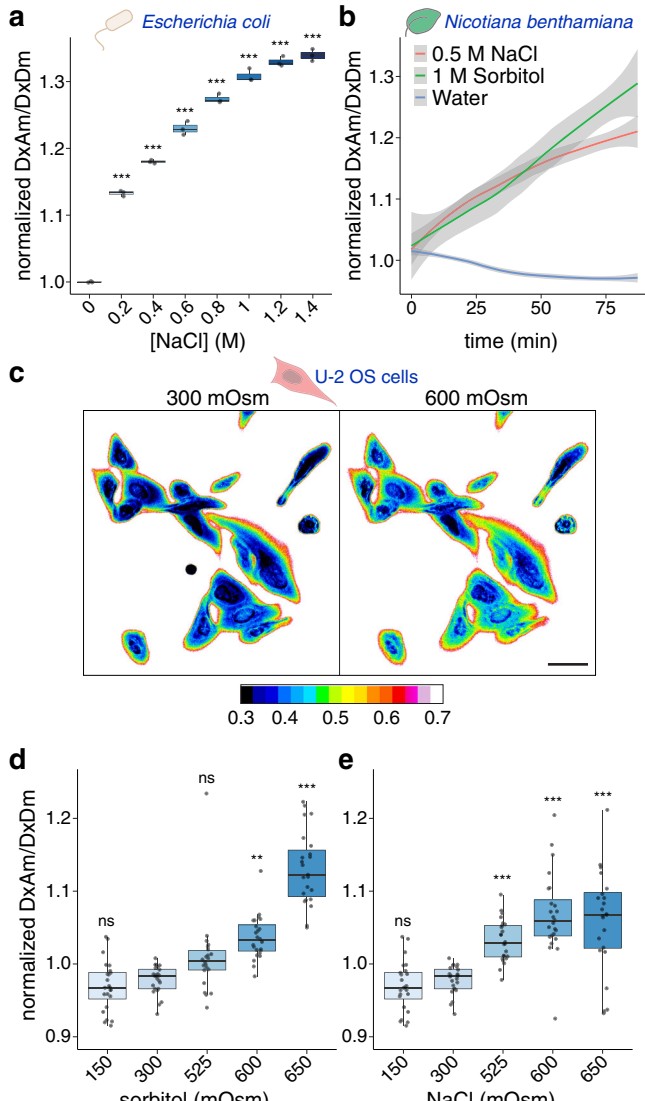

**Fig. 5 SED1 tracks changes in osmolarity of a wide set of organisms.** **a** Normalized FRET ratio (DxAm/DxDm) of live SED1-expressing *Escherichia coli* cells treated with different concentrations of NaCl. $n = 3$ independent experiments. One-way ANOVA. $*p < 0.05$, $**p < 0.01$, $***p < 0.001$. Boxes represent 25th–75th percentile (line at median) with whiskers at 1.5*IQR. **b** Normalized FRET ratio (DxAm/DxDm) time course of *Nicotiana benthamiana* leaf discs transiently expressing SED1, treated with either water, 0.5 M NaCl, or 1 M sorbitol. $n = 7$–11 leaf discs. Mean ± SEM. One-way ANOVA. **c** Ratiometric image of live SED1-expressing U-2 OS cells at 300 mOsm (isosmotic) or 600 mOsm (hyperosmotic) treated with sorbitol. Scale bar = 50 μm. Calibration bar represents the normalized FRET ratio (DxAm/DxDm). **d** Normalized FRET ratio of SED1-expressing U-2 OS cells exposed to different osmotic treatments with sorbitol. $n = 5$ cells, 5 regions of interest per cell. One-way ANOVA. $*p < 0.05$, $**p < 0.01$, $***p < 0.001$. Boxes represent 25th–75th percentile (line at median) with whiskers at 1.5*IQR. **e** Normalized FRET ratio of SED1-expressing U-2 OS cells exposed to different osmotic treatments with NaCl. $n = 5$ cells, 5 regions of interest per cell. One-way ANOVA. $*p < 0.05$, $**p < 0.01$, $***p < 0.001$. Boxes represent 25th–75th percentile (line at median) with whiskers at 1.5*IQR. Source data are provided as a Source Data file.

0.5 M NaCl. Contrary to *N. benthamiana*, we did not observe hyperosmotic-dependent FRET ratio changes (Supplementary Fig. 9). Since AtLEA4-5 - the sensory domain of SED1 - is an Arabidopsis protein, the lack of response could be the result of its

interaction with endogenous binding partners and/or posttranslational modifications. In agreement with the latter hypothesis, it was recently reported that LEA proteins are hyperphosphorylated at almost every serine, threonine, and tyrosine residue in Arabidopsis[36]. The introduction of several negative charges throughout the protein likely prevents hyperosmotic stress-induced compaction.

We further tested SED1 in human cells. To do so, we stably introduced SED1 into human U-2 OS cells and measured the SED1 FRET signal in response to sorbitol and NaCl treatments at different osmolarities using live-cell confocal microscopy. We observed that both treatments induced an increased FRET ratio immediately after the addition of the solution (Fig. 5c–e). The increased fluorescence of the acceptor and decreased fluorescence of the donor after the treatments, along with the acceptor photobleaching control, confirmed the expected FRET behavior (Supplementary Fig. 10). These data demonstrate that SED1 is responsive in live human cells, complementing our observations in bacteria, yeast and plant cells (See Supplementary Fig. 11 for baseline FRET ratio comparison of the different organisms).

In conclusion, we showed that SED1 is a versatile, genetically encoded optical tool that can be used to dynamically track the response to osmotic stress of living cells from various organisms in an inherently quantitative manner. This opens new avenues to investigate the poorly understood impact of environmental perturbations on the regulation of cellular function.

## Discussion

Recent progress in the characterization of the molecular and cellular functions of IDRs has revolutionized our understanding of cell biology. Protein domains that lack a defined and stable structure perform important functions including signaling, transcriptional and translational regulation, stress protection, and control of enzymatic function[37–39]. Dysregulation of IDR-containing proteins (like TP53, alpha-synuclein, and TDP-43) often results in disease[40]. IDRs are key players in the process of liquid-liquid phase separation (LLPS), which is thought to mediate the formation of intracellular membraneless compartments such as the nucleolus[41]. Notably, the dynamic conformational structure of IDRs can be modulated by interaction with binding partners, posttranslational modifications, or changes in the chemical environment of the solution[17,20,25]. Despite this significant progress in understanding the functions of IDRs, their potential for building molecular tools such as biosensors has been largely overlooked. In this work, we leveraged the unique features of IDRs to develop a highly sensitive biosensor that exploits the flexible nature and sensitivity to osmolarity changes of a plant IDR. Because of the unique capabilities mentioned above and the high prevalence of IDRs in the proteomes of organisms across all kingdoms of life, we anticipate that this work will pave the way for using IDRs to develop other cutting-edge molecular tools.

The first generation of genetically encoded biosensors often come with a handful of limitations[42]. Subsequent biosensor optimization focuses on larger dynamic range, improved specificity, higher brightness, and fewer undesired side-effects[43]. It is very likely that future rounds of SED1 improvement would be designed specifically for each kind of organism where it is to be used. The use of computational solution space scanning will likely aid in these efforts as the role of individual amino acid variants can be tested.

While SED1 can dynamically monitor the effects of osmotic stress in Escherichia coli, Saccharomyces cerevisiae, Nicotiana benthamiana, and human cells, the main caveat of SED1 is the lack of responsiveness in the model plant Arabidopsis thaliana. Since the sensory domain of SED1 (AtLEA4-5) is an Arabidopsis

protein, we hypothesize that the environmentally driven conformational changes might be affected by interaction with its endogenous binding partners, or by posttranslational modifications, as recently shown for other LEA proteins[36]. Tuning SED1 sequence to make it sufficiently different from endogenous AtLEA4-5 while maintaining its environmental responsiveness would be the next step in the refinement of this biosensor for Arabidopsis.

SED1 has a larger dynamic range than the macromolecular crowding biosensor (CS)[14]. The CS sensory domain is a synthetic peptide composed of two α-helices in a hinge-like topology. AtLEA4-5 is a naturally occurring intrinsically disordered protein that lacks secondary structure, yet exhibits a more dramatic response to increased macromolecular crowding than CS both in vitro and in vivo. In addition, SED1 offers advantages over other existing fluorescent crowding reporters due to its genetically encoded character. Gnutt et al developed a random coil polyethylene glycol FRET-based reporter to quantify crowding changes in vitro and in mammalian cells subjected to osmotic stress[15]. Köning et al labeled the intrinsically disordered prothymosin α (ProTα) and used it as a crowding indicator in mammalian cells[21]. Both reporters rely on in vitro labeling and subsequent microinjection of the labeled construct into the cell. While this presents an advantage in terms of the fluorescent properties of the reporters and good control over their intracellular concentration, the process requires expertise in both labeling and microinjection, creating a barrier for their wide use and throughput capabilities.

The disorder levels and amino acid composition of AtLEA4-5 are not sufficient to explain the dramatic response to changing cellular conditions, since all the different scrambled versions we tested had a decreased ability to compact (Fig. 1f and Fig. 2a). Therefore, the primary sequence, and the intramolecular interactions they facilitate in AtLEA4-5, form the molecular underpinnings for the stress-induced compaction observed in vivo, supporting our previous evidence in vitro[25]. Indeed, the sensitivity of IDRs to their physical–chemical environment has been shown to be sequence dependent, and might regulate their function, as proposed previously[20]. The functional regulation of disordered domains by environmental factors could have enormous implications, especially for organisms with high content of IDRs in their proteome. This work contributes to a better understanding of how the primary sequence of disordered regions accounts for their sensitivity to the physical–chemical properties of the environment in cells.

Single budding yeast cells expressing SED1 displayed different FRET levels under non-stress conditions and may be due to variation in protein concentration and overall macromolecular crowding in cells. Additional variation in the population was observed upon hyperosmotic shock where cells with larger vacuoles showed a smaller change in FRET after treatment. Vacuoles serve as a reservoir of water and allow cells to lose water under hyperosmotic stress without dramatically changing the concentration of solutes in the cytoplasm. Whether other vacuole-containing organisms such as plants display similar variation in SED1 FRET levels remains to be studied and opens the possibility for understanding how cells cope with different water availability through the use of intracellular compartments.

The use of SED1 to monitor osmotic variations in living cells has the potential to reveal fundamental aspects of cell biology. SED1 might be used to (1) dynamically track the macromolecular crowding of individual cells during perception, response, and acclimation to osmotic stress; (2) screen for mutants disrupted in the sensing and response mechanisms to osmotic shock; (3) test whether other kinds of stressors induce intracellular osmotic variation; (4) generate osmolarity and/or macromolecular

crowding maps of different cell types of multicellular organisms. This has the potential to revolutionize our understanding of the biological processes that enable desiccation survival, extreme salt tolerance, and rehydration. The ability of SED1 to work in evolutionarily distant organisms means that these processes can be studied across the tree of life to broaden our understanding of the ways in which water impacts life on Earth.

## Methods

**All-atom simulation.** Simulations of AtLEA4-5 protein, its scrambles, and other IDRs were done using Solution Space Scanning[19], an all-atom Monte Carlo simulation method based on the ABSINTH force field[44,45] that has been previously described[19]. Briefly, the Hamitonian function to be evaluated in each step can be written as the following representation.

$$E_{total} = W_{solv} + U_{Lj} + W_{el} + U_{corr}$$

Here, $W_{solv}$ is the energy describing the interaction between the protein surface and the surrounding solution. By changing the $W_{solv}$ term, we can alter this interaction and sample a protein's conformations in different solution conditions.

For each combination of solution condition and protein (AtLEA4-5 and each of its sequence scrambles), we ran five independent simulations consisting of $5 \times 10^7$ Monte Carlo steps (following $1 \times 10^7$ steps of equilibration) starting from random conformations to ensure proper sampling. Protein conformations were written out every 12,500 steps. The dataset of 70 other IDRs shown in Fig. 2A was obtained using the same methods, is publicly available on https://github.com/sukeniklab/HiddenSensitivity[20], and has been previously described[20]. We analyzed the average radii of gyration of the simulated conformation ensembles using the MDTraj python library[46]. Standard deviations were calculated based on the average of five individual repeats. Each radius of gyration was then normalized based on the most expanding solution to highlight solution sensitivity.

**Transgene constructs.** pDRFLIP38 backbone was used for biosensors yeast expression[47]. This plasmid contains the constitutive promoter pPMA1, and was provided by Dr. Alexander M. Jones. The vector was digested with XbaI (NEB) and EcoRI (NEB) to clone the open reading frames (ORFs) of mCreluan3, AtLEA4-5, and Citrine downstream of the pPMA1 promoter. The biosensor construct was cloned using the Gibson Assembly cloning method (NEB) by mixing the XbaI-EcoRI-digested pDRFLIP38 with the PCR-amplified ORFs containing overlapping ends. The ORFs of the other fluorescent proteins (t7.eCFP.t9, Aphrodite.t9, t7.TFP.t9, mTFP.t9, Cerulean, edCerulean, edCitrine, edAphrodite.t9) used in this study were cloned in the same way. The sensory domains tested (AtLEA4-2, ABP, CS, N-AtLEA4-5, C-AtLEA4-5, Scramble-1, Scramble-2, Scramble-3, Scramble-4, Scramble-5) were cloned between mCerulean3 and Citrine ORFs. To do this, pDRFLIP38-AtLEA4-5 was digested with SacI and BglII to remove the AtLEA4-5 ORF. The digested plasmid was mixed with the different PCR-amplified sensory domains-ORFs containing overlapping ends using the Gibson Assembly method (NEB). AtLEA4-2, AtLEA4-5, N-AtLEA4-5, and C-AtLEA4-5 ORFs were amplified from pTrc99A-AtLEA4-2 and pTrc99A-AtLEA4-5 plasmids provided by Dr. Alejandra A. Covarrubias[25]. ABP ORF was amplified from pGW1araF.Ec plasmid provided by Dr. Wolf B. Frommer[26]. CS ORF was amplified from Cr1-pRSET-A provided by Dr. Arnold Boersma[14]. Scrambled versions were randomly designed using the Scrambler tool of PeptideNexus (https://peptidenexus.com/). Scrambles were chosen based on disorder propensity, α-helix prediction (AGADIR web server http://agadir.crg.es/) and charge mixing (Kappa value)[27]. All AtLEA4-5 Scrambled ORFs were synthesized as gene fragments (GenScript).

For bacterial expression, pDEST-HisMBP backbone was used (Addgene #11085). This plasmid contains the Tac IPTG-inducible promoter for protein expression with a N-terminal 6x His tag and MBP tag. The full SED1 ORF was cloned into pENTR-D-TOPO (Thermo Fisher Scientific). Recombination of pENTR-D-TOPO-SED1 and pDEST-HisMBP was done using Gateway technology to produce pDEST-HisMBP-SED1. The same strategy was followed for the full CS ORF to produce pDEST-HisMBP-CS.

pGPTVII-U-MCaMP6s binary vector was used for expression in plant cells[48]. This plasmid contains the AtUBQ10 promoter, and was provided by Dr. Cindy Ast. The plasmid was digested with SpeI (NEB) and XmaI (NEB). SED1 was cloned using the Gibson Assembly cloning method (NEB) by mixing the SpeI-XmaI-digested pGPTVII-Bar-U-MCaMP6s with the PCR-amplified SED1 ORF containing overlapping ends. The nuclear localization signal was added in the forward primer (ATGCTGCAGCCTAAGAAGAAGAGAAAGGTTGGAGGG).

**Transgene expression.** The constructs indicated in the main text were transformed into *Saccharomyces cerevisiae* protease-deficient yeast strain (BJ5465 lacking Pep4 and Prb1) using the lithium acetate transformation method[49]. Transformed colonies were selected in plates containing 6.8 g/L YNB media (Sigma-Aldrich) supplemented with 5 g/L glucose and 1.92 g/L synthetic drop-out medium without uracil (Sigma-Aldrich). Positive clones were confirmed by colony PCR. SED1 was also transformed into wild type and *hog1Δ::G418* and *pbs2Δ::G418*

mutant backgrounds of the *Saccharomyces cerevisiae* BY4742 strain (provided by Dr. Hugo Tapia). Transformation and selection were done as described above.

pDEST-HisMBP-SED1 was transformed into *Escherichia coli* BL21 (DE3) strain using the standard heat shock transformation protocol. Transformed colonies were selected in plates containing LB media supplemented with ampicillin (100 µg/mL). Positive clones were confirmed by colony PCR. The same strategy was followed for pDEST-HisMBP-CS.

pGPTVII-nlsSED1 was transformed into *Agrobacterium tumefaciens* GV3101 (pSoup) strain using the electroporation method. Transformed colonies were selected in plates containing LB media supplemented with gentamicin (50 µg/mL), kanamycin (50 µg/mL), and tetracycline (2 µg/mL). Positive clones were confirmed by colony PCR. For transient expression in *Nicotiana benthamiana*, the positive *Agrobacterium tumefaciens* clones and p19 strain were co-transfected in large *Nicotiana benthamiana* leaves and incubated for 5 days before measurements[50]. For stable expression in *Arabidopsis thaliana*, four pots of 30 days-after-sowing (flowering) Col-0 plants were transformed with the positive *Agrobacterium tumefaciens* clones using the floral dip method[51]. T1 transformed seeds were selected in MS media containing DL-Phosphinothricin herbicide. Three independent T3 homozygous plants were selected for imaging.

**Fluorescence analysis of live *Saccharomyces cerevisiae* cells.** 5 mL of yeast cells expressing the indicated constructs (see main text) were grown at 30 °C in liquid YNB media (6.8 g/L) (Sigma-Aldrich) supplemented with 5 g/L glucose and 1.92 g/L synthetic drop-out medium without uracil (Sigma-Aldrich) until OD600 ~ 1–2. Cells were centrifuged and washed twice with 50 mM MES, pH 6 and resuspended in 5 mL of the same buffer. 50 µL of the cell suspension was loaded into individual wells of a 96-well black F-bottom clear microplate (Greiner). 150 µL of treatment solution (see main text) was added to the cell suspension, mixing was performed by pipetting up and down, and the fluorescence was measured immediately after. Fluorescence readings were acquired using a Safire fluorimeter (Tecan) for donor fluorophore (mCerulean3 excitation 433 nm, mCerulean3 emission 480 nm, abbreviated DxDm), acceptor fluorophore (Citrine excitation 510 nm, Citrine emission 525 nm, abbreviated AxAm), and energy transfer from donor to acceptor (mCerulean3 excitation 433 nm, Citrine emission 525 nm, abbreviated DxAm). Fluorescence emission scans from 460 nm to 550 nm (step size 5 nm) with an excitation wavelength of 433 nm were acquired. For all fluorescence measurements, bandwidth was set to 5 nm (7.5 nm for the emission scan), number of flashes was 10, integration time was 40 µs, and gain was 100. For time course measurements, the 96-well plate was kept inside the plate reader for the duration of the experiment. Measurements were acquired every 60 s for a period of 120 to 150 min. Shake (linear) duration was set to 3 s before every measurement. Nine independent measurements were acquired for each treatment and construct. Experiments were repeated three times.

**Fluorescence analysis of live *Escherichia coli* cells.** 3 mL of SED1-expressing *Escherichia coli* culture was grown at 37 °C in liquid LB supplemented with ampicillin to OD600 ~ 1–2. No IPTG induction was needed since the fluorescence obtained from the leaking expression of the Tac promoter was sufficient for measurements. Cells were centrifuged and washed twice with 50 mM MES, pH 6 and resuspended in 3 mL of the same buffer. 50 µL of the cell suspension was loaded into individual wells of a 96-well black F-bottom clear microplate (Greiner). 150 µL of treatment solution (see main text) was added to the cell suspension, mixing was performed by pipetting up and down, and the fluorescence was measured immediately after. Fluorescence readings were acquired using a Safire fluorimeter (Tecan) for donor fluorophore (mCerulean3 excitation 433 nm, mCerulean3 emission 480 nm, abbreviated DxDm), acceptor fluorophore (Citrine excitation 510 nm, Citrine emission 525 nm, abbreviated AxAm), and energy transfer from donor to acceptor (mCerulean3 excitation 433 nm, Citrine emission 525 nm, abbreviated DxAm). Fluorescence emission scans from 460 nm to 550 nm (step size 5 nm) with an excitation wavelength of 433 nm were acquired. For all fluorescence measurements, bandwidth was set to 7.5 nm, number of flashes was 10, integration time was 40 µs, and gain was 100. Three independent measurements were acquired for each treatment and construct.

**Fluorescence analysis of live *Nicotiana benthamiana* cells.** *Nicotiana benthamiana* leaf discs were obtained from leaves transiently expressing SED1 and loaded into individual wells of a 96-well black F-bottom clear microplate (Greiner). 5 µL of water was added to the bottom of the well before loading the leaf discs to facilitate a flattened distribution of the leaf to the bottom of the well. 20 µL of the indicated solution (see main text) was added to the top of each leaf disc and fluorescence was measured immediately after. The 96-well plate was kept inside the plate reader for the duration of the experiment. Measurements were acquired every 180 s for a period of 90 min. Shake (linear) duration was set to 3 s before every measurement. Fluorescence readings were acquired using a Safire fluorimeter (Tecan) for donor fluorophore (mCerulean3 excitation 433 nm, mCerulean3 emission 480 nm, abbreviated DxDm), acceptor fluorophore (Citrine excitation 510 nm, Citrine emission 525 nm, abbreviated AxAm), and energy transfer from donor to acceptor (mCerulean3 excitation 433 nm, Citrine emission 525 nm, abbreviated DxAm). For all fluorescence measurements, bandwidth was set to

5 nm, number of flashes was 10, integration time was 40 μs, and gain was 100. Three independent experiments were carried out with 7-11 leaf discs for each treatment.

**Saccharomyces cerevisiae fluorescence microscopy.** 5 mL of yeast cells expressing the indicated constructs (see main text) were grown at 30 °C in liquid YNB media (6.8 g/L) (Sigma-Aldrich) supplemented with 5 g/L glucose and 1.92 g/L synthetic drop-out medium without uracil (Sigma-Aldrich) until OD600 ~ 1–2. Cells were centrifuged and washed twice with 50 mM MES, pH 6 and resuspended in 5 mL the same buffer. 100 μL of the cell suspension was loaded into a μ-Slide 8-well Ibidi chamber (Ibidi GmbH) and mixed with 100 μL of the treatment solution (2X) to reach the desired final concentration (see main text). Imaging was done immediately after the treatment.

Imaging was performed with a Leica TCS SP8 laser scanning confocal microscope with LASX software. For intensity-based measurements a 63x/1.4 NA oil HCX PL APO immersion objective was used for all the experiments. Thirty Z-stack images with steps of 0.3 μm (system optimized) were captured. Three kinds of images were acquired in the sequential mode for each experiment: donor emission with donor excitation (DxDm; donor channel), acceptor emission with donor excitation (DxAm; FRET channel), and acceptor emission with acceptor excitation (AxAm; acceptor only channel). For DxDm, excitation = 440 nm; emission = 450–500 nm. For DxAm, excitation = 440 nm; emission = 525–550 nm. For AxAm, excitation = 514 nm; emission = 525–550 nm. Laser power was set between 2 and 5% and detector gain was set to 80. At least three different fields of view were acquired for each strain/treatment. Fluorescence emission was detected by HyD detectors. A line average of eight was used for all the experiments.

For 1,6-hexanediol treatment, only AxAm was followed. Cells were first treated with 0.5 M NaCl and subsequently incubated with 10% (w/v) 1,6-hexanediol. Imaging was done at the parameters indicated above.

FLIM-FRET experiments were carried out on yeast strains containing AtLEA4-5-mCerulean3 (donor-only) and SED1 (donor and acceptor: mCerulean3 and Citrine). FLIM was measured using a Leica TCS SP8 FALCON confocal microscope with LASX software. A 93x/1.3 NA glycerin immersion objective was used for all experiments. Excitation of the donor fluorophore was done at 440 nm with a diode pulsed laser at 40 MHz repetition rate. Emission from 450-515 nm was collected with a HyD SMD hybrid detector. Laser power was adjusted to obtain a maximum of ~1 photon per laser pulse, and 20 frames were integrated. The pixel frame size was set to 512, which gave a pixel size of 0.24 μm.

**Saccharomyces cerevisiae image analysis.** For intensity-based measurements, the images were analyzed using Fiji (http://fiji.sc/). Sum of Z-stacks and background subtraction were performed for DxAm, DxDm, and AxAm images. Regions of interest corresponding to individual cells were selected and ratios of DxAm/DxDm were calculated and normalized to the no-treatment mean. To generate the ratiometric image, Gaussian blur of 1 was applied to all images. AxAm was used to create a mask. DxAm/DxDm ratio image was multiplied by the mask image, divided by 255 and set to the same range for all images. The background was manually set to white pixels for clarity.

FLIM measurements were analyzed using LASX software. FRET efficiency was calculated for the entire image by comparing the FLIM values obtained for the mCerulian3 donor fluorophore in the presence and absence of the Citrine acceptor fluorophore. First, the average fluorescence lifetime of the donor was determined by fitting every image with a single exponential component. The resultant lifetime value was used as the Unquenched Donor Lifetime parameter to calculate the FRET efficiency by applying a mono-exponential decay model to fit the experimental decays. For individual SED1-expressing cells, a region of interest was selected for every cell and lifetime was calculated by fitting the fluorescence decay with a single exponential model. The vacuolar ratio was calculated as the ratio between vacuole area and total area per cell. Vacuole area and total area per cell were obtained from the fluorescence intensity images using Fiji.

**Protein purification.** 2 L of LB supplemented with ampicillin (100 μg/mL) was inoculated with 20 mL/L of a saturated culture of cells containing pDEST-HisMBP-SED1. The culture was grown at 37 °C, shaking at 225 rpm, to OD600 ~ 0.6. At this point, recombinant protein was induced with 1 mM IPTG (BIOSYNTH International) and the culture was transferred to 16 °C and 250 rpm for 20 h. Cells were collected and lysed in lysis buffer (50 mM $NaH_2PO_4$, pH 8, 0.5 M NaCl) by sonication on ice using a Q700 sonicator (Qsonica), and the extract was clarified by centrifugation. The His-tagged recombinant fusion protein was separated by affinity chromatography using Ni-NTA beads (Qiagen) and eluted with 50 mM $NaH_2PO_4$, pH 8, 0.5 M NaCl, 250 mM imidazole. To remove the His-MBP tag, the recombinant protein was incubated at 4 °C overnight with TEV protease. Tag-free recombinant SED1 was separated by size-exclusion chromatography in a ÄKTA purification system (Cytiva) in 20 mM sodium phosphate, pH 7.4, 100 mM NaCl. The purity of recombinant SED1 was confirmed by SDS-PAGE. The same strategy was followed for pDEST-HisMBP-CS.

**Solution preparation and specifics.** Solutes were purchased from Alfa Aesar (Sarcosine, PEG200, PEG400, PEG1500, PEG2000, PEG4000, PEG6000, PEG8000, PEG10000), VWR (D-Sorbitol), GE Healthcare (Ficoll), TCI (D-(+)-Trehalose Dihydrate, Trimethylamine N-Oxide Dihydrate (TMAO)), Thermo Scientific (Guanidine Hydrochloride), Acros Organics (Betaine Monohydrate, and Fisher BioReagents (Ethylene Glycol, Glycerol, Glycine, Magnesium Chloride Hexahydrate, Potassium Chloride, Sodium Chloride, Sucrose, Urea), and used without further purification. Stock solutions were made by mixing the solute with 20 mM sodium phosphate buffer, pH 7.4, with the addition of 100 mM NaCl except for NaCl and KCl solutions, which were initially free of additional salt. The same buffer was used for all dilutions.

**Fluorescence analysis of purified recombinant proteins.** FRET experiments were conducted in black plastic 96-well plates (Nunc) using a CLARIOstar plate reader (BMG LABTECH). Buffer, stock solution, and purified protein solution were mixed in each well to reach a volume of 150 μL containing the desired concentrations of the solute and the FRET construct, with a final concentration of 0.8 μM protein. Fluorescence measurements were taken from above, at a focal height of 5.7 mm, with gain fixed at 1020 for all samples. For each construct, 24 replicates were performed in neat buffer containing NaCl, 12 replicates were performed in neat buffer not containing NaCl, and two repeats were performed in every other solution condition. Fluorescence spectra were obtained for each construct in each solution condition by exciting the sample in a 10-nm band centered at 433 nm, with a dichroic at 446.5 nm, and measuring fluorescence emission from 460 to 600 nm, averaging over a 10 nm window moved at intervals of 1 nm. The ratio of acceptor to donor intensity (DxAm/DxDm) was calculated by dividing the total measured fluorescence intensities from 500 to 600 nm by the total measured fluorescence intensities from 460 to 499 nm.

**U-2 OS cell culture.** All U-2 OS (ATCC HTB-96) and HEK-293T (ATCC CRL-3216) cell lines used in this study were cultured at 37 °C in 5% $CO_2$ in high-glucose DMEM (GE Healthcare) supplemented with 10% FBS (Atlanta Biologicals), 1 mM sodium pyruvate (Gibco), 2 mM L-glutamine (Gemini Biosciences), 1x MEM non-essential amino acids (Gibco), 40 U/ml penicillin and 40 μg/ml streptomycin (Gemini Biosciences). Stable U-2 OS SED1-expressing cell lines were generated by lentiviral transduction. To produce lentiviral particles, the SED1 construct was first subcloned into EcoRV-HF (NEB)-digested pLenti-CMV Puro DEST (Addgene #17452) using the NEBuilder HiFi DNA Assembly master mix (NEB), and then transfected into HEK-293T cells together with pMD2.G (Addgene #12259) and psPAX2 (Addgene #12260). Virus was harvested 48 h after transfection, filtered through non-binding 45 μm syringe filters (Pall Corporation) and used to transduce U-2 OS cells. After 24 h, the virus-containing medium was removed and replaced with selection medium containing 2 μg/ml Puromycin (Sigma–Aldrich). After 7 days of selection, single-cell clones were derived by sorting for the top ~60% fluorescent cells using a Sony SH800 flow cytometer. Two individual clones were randomly selected for further use.

**U-2 OS sample preparation.** U-2 OS cells expressing SED1 were cultured in Corning treated flasks with Dulbecco's modified Eagle's medium (DME:F-12 1X from Hyclone Cat No SH30023.01) supplemented with 10% FBS (Gibco REF 16000-044) and 1% penicillin/streptomycin (Gibco REF 15140-122). Cells were incubated at 37 °C and 5% $CO_2$. Sorbitol (VWR CAS 50-70-4) and NaCl (Fisher Bioreagents CAS 7647-14-5) stock solutions of 3 M and 5 M respectively were prepared by dissolving the corresponding amounts of sorbitol or NaCl in autoclaved DI water and filtering using a 0.2 μm filter. The solutions used for perturbations were obtained by diluting the stock solutions with autoclaved DI water.

Prior to imaging, 13,000 cells were plated in a μ-Plate 96-well black treated imaging plate (Ibidi) and allowed to adhere overnight (~16 h) before perturbations. Cells were stained with DAPI (Thermo). To prepare the stain, a 14.3 mM DAPI stock dissolved in DI water was diluted to a final concentration of 300 μM with complete media. The media from the cells was aspirated and DAPI-containing media was added to the cells, which were then incubated for 15 min at 37 °C and 5% $CO_2$. After the incubation period, the cells were rinsed twice with PBS and 200 μL of PBS was added.

**U-2 OS fluorescence microscopy.** Imaging was done on a Zeiss epifluorescent microscope using a 40 × 0.9 NA dry objective. Excitation was done with a Colibri LED excitation module and data were collected on dual Hamamatsu Flash v3 sCMOS cameras. The cells were imaged at room temperature before and less than 1 min following perturbation with 300 ms exposure times. Imaging was done by exciting DAPI (385 nm) under donor excitation (Dx, 430 nm) or acceptor excitation (Ax, 511 nm). Emitted light was passed on to the camera using a triple bandpass dichroic (467/24, 555/25, 687/145). When measuring FRET, emitted light was split into two channels using a downstream beamsplitter with a 520 nm cutoff. For each perturbation, the cells were focused using the DAPI channel, and imaged with two channels using Dx, in one channel using Ax. The final osmolarities that were used for the perturbations were: 150 mOsm, 300 mOsm (isosmotic), 525 mOsm, 600 mOsm, and 650 mOsm with sorbitol or NaCl as the osmotic agents. From each well in the 96-well plate, 4-5 cells were analyzed. Each

perturbation was replicated at least 3 times in a single plate, and the data reported are combined from at least two plates prepared on different days.

**U-2 OS image analysis**. The images were analyzed using ImageJ. For each cell, 5 ROIs were selected: (1) background ROI, located where no cells were present, to measure any background changes that may have occurred due to media changes; (2–5) four ROIs in the cytoplasm of each cell. For each ROI, the background signal was subtracted, and average intensity values were reported in four channels: (a) donor emission under donor excitation (DxDm), (b) acceptor emission under donor excitation (DxAm), (c) acceptor emission under acceptor excitation (AxAm), and (d) DAPI emission under DAPI excitation. To correct for donor bleedthrough, cells were plated and stained as previously mentioned. Cells were imaged, the acceptor was photobleached under prolonged direct acceptor excitation, and the cells were imaged again. ROIs of all the cells present in the plane of view were measured. A correlation plot of donor emission against acceptor emission was generated to determine percent bleedthrough, as shown in Supplementary Fig. 10.

**Quantification and statistical analysis**. Data were analyzed with one-way ANOVA or two-way ANOVA for all experiments with more than two samples, as indicated in the figure legends, with Tukey's multiple comparison test. For experiments with two samples, data were analyzed using unpaired Student's $t$ test. Symbols *, **, and *** indicate $p$-values < 0.05, 0.01, and 0.001, respectively, unless specified differently in the figure legends.

**Reporting summary**. Further information on research design is available in the Nature Research Reporting Summary linked to this article.

## Data availability

The authors declare that data supporting the findings of this study are available within the paper and its supplementary information files. Source data are provided with this paper.

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

## Acknowledgements

The authors thank members of the Dinneny lab for critical review of the manuscript; members of the Carnegie-Stanford Intrinsically Disordered Proteins Scientific Interest Group (IDPSIG) for helpful discussions and for fostering important collaborations for this manuscript; Heather Cartwright for technical assistance in confocal microscopy; Shouling Xu for assistance with FPLC; Dr. Hugo Tapia for providing wild type BY4742 strain as well as *hog1Δ::G418* and *pbs2Δ::G418* mutants; Dr. Arnold Boersma for providing Cr1-pRSET-A plasmid; Dr. Wolf B. Frommer for providing pGW1araF.Ec plasmid; and Dr. Cindy Ast for providing pGPTVII-Bar-U-MCaMP6s plasmid.

C.L.C.V. was a Latin American Fellow in the Biomedical Sciences, supported by the Pew Charitable Trusts. The research of J.R.D. was supported in part by a Faculty Scholar grant from Howard Hughes Medical Institute and the Simons Foundation (55108515). This work was supported by UNAM-PAPIIT IA209920. We gratefully acknowledge computing time on the MERCED cluster at UC Merced, funded by NSF grant No. ACI-1429783, and on the XSEDE computational infrastructure framework, grant No. TG-MCB190103 to S.S., supported by NSF grant No. ACI-1548562. S.S., D.M., and F.Y. were supported by the NIH under award R35GM137926 to S.S.; D.M. and K.G. were supported by a fellowship from NSF-CREST Center for Cellular and Biomolecular Machines at UC Merced, grant No. NSF-HRD-1547848.

## Author contributions

Conceptualization: C.L.C.V., A.A.C., and J.R.D.; Methodology: C.L.C.V., J.A.N.B., S.S., and J.R.D.; Software: F.Y., and S.S.; Formal Analysis: C.L.C.V., T.V., F.Y., and S.S.; Investigation: C.L.C.V., T.V., K.G., H.B.S., F.Y., D.M., J.A.N.B., D.C.A., A.D., and L.W.; Resources: A.M.J.; Writing—Original Draft: C.L.C.V. and J.R.D.; Writing—Review & Editing: C.L.C.V., T.V., K.G., H.B.S., D.M., J.A.N.B., A.D., A.M.J., A.A.C., S.S., J.R.D.; Visualization: C.L.C.V., T.V., and S.S., Supervision: C.L.C.V., A.A.C., S.S., and J.R.D.; Funding Acquisition: C.L.C.V., A.A.C., S.S., and J.R.D.

## Competing interests

The authors declare no competing interests.
