## [Peer Review File · Nature Communications]

Reviewers' Comments:

Reviewer #1:

Remarks to the Author:

Cuevas-Velasquez et al. describe a new fluorescent biosensor based on the environmental sensitivity of intrinsically disordered regions, and show that it can be used to monitor osmotic stress in bacteria, yeast and some plants and cultured human cells. The sensor shows a significant improvement relative to existing crowding sensors, and represents a new type of biosensor design that will likely lead to many follow-up studies. The manuscript presents a multifaceted characterization of the sensor using both *in silico*, *in vitro* and *in vivo* experiments and represents both a significant advance in fundamental understanding (vacuole-size correlates with stress resistance) and provides a useful new research tool. There are many avenues that could be pursued further, but in most cases these are better left for follow-up studies. There are, however, a few topics that are so central to this paper that I think it should be addressed before publication: It is unclear exactly what the biosensor measures, what the dynamic range is, and whether the reported FRET values can be compared between different experiments. These questions could either be addressed through additional experiments, analysis or discussion depending on the authors' judgement.

Major points:

1. What does the sensor respond to? As the authors note, osmotic stress introduces several changes to the chemical environment at once, and it is important to clarify which are responsible for the compaction. In Fig. 2B it is convincingly shown that crowding is part of the answer. Another factor that is likely to have an impact is ionic strength as the compaction of IDPs are highly sensitive to electrostatic interactions. It would be nice to know how the purified reporter responds to ionic strength and possibly different solutes (using a similar setup to Ref. 19). Also, does crowding effect observed in Fig. 2B explain the entire change seen in the osmotic response in cell experiment or is something missing? (It is difficult to compare the values in Fig. 2B to the remaining FRET reporter values as they are reported in different units).
2. Can the sensor be saturated? Fig. 1B shows a plateau in the FRET ratio at high osmolyte values. Does this represent an intrinsic plateau in the dynamic range of the sensor or a plateauing in the osmotic response of the cells. One way to answer this would be to clarify whether there are signs of saturation in the experiments performed with the purified sensor.
3. Analysis of FRET data. Through-out the manuscript the fluorescent read-out as "normalized Dx_{Am}/Dx_{Dm} ". As I understand the methods description, these are the raw fluorescence values and are not corrected for back-scattering, direct excitation of the acceptor and bleed-through of the donor emission into the acceptor signal (which will be significant for these FPs). If I understand this correctly, then this means that a change from "normalized Dx_{Am}/Dx_{Dm} " 1 to 1.3 will represent something different for different experiments, and mean that the read-out from the biosensor should not be directly compared except for very similar experiments. I think the authors should consider whether it would be possible to correct the fluorescence signals to get values that are more comparable across various setups. Also, the normalization may be needed to get clean comparisons across a range of experiments, but removes potentially useful information. I think you should consider either presenting unnormalized data, or alternatively present the base line FRET ratio for each normalized value to allow the reader to judge which normalized values are comparable.
4. It is a little concerning and inconvenient that the sensor partitions to stress granules in yeast (Fig. S5c). The 10% 1-6-haxanediol control shows that the increase in FRET is not due to this partitioning, however the FRET ratio is significantly higher in the presence of 1,1-hd (Fig. S5d). It would be beneficial if this was discussed more fully, as this potentially raises the question of what environment the sensor samples. Also, it would be useful if the authors could state whether something similar happens in all the cell types they have tested, or whether it is specific to yeast.
5. Clear discussion of limitations. The discussion currently focuses on the potential applications of the sensor, and largely omits discussion of the limitations of the sensor. I think your readers will benefit from a paragraph dedicated to the limitations of the sensor, where you clearly lay out what

can and cannot be concluded (drawing on the points above). I would like to point out that I think that this paper clearly merits publication in a good journal, and that an explicit recognition of limitation of the study does not detract from this.

Minor points:

6. The authors use the term "plethora" in both the introduction and discussion. In my minds, this term has negative connotations and thus likely do not reflect what the authors mean, and you may wish to consider a different wording.
7. p. 6 "The FRET ratio change was significantly smaller when we tested a globular protein (arabinose-binding protein, ABP) as a reference" - I do not understand this control. Maybe you want to elaborate on why this protein was chosen.
8. The FRET efficiencies from FLIM should probably not be reported with 4 significant digits.
9. How were the scrambled sequences exactly constructed? By manually or using an algorithm? Were other parameters also considered e.g. the degree of charge mixing? (Can be quantified from the CIDER web server).
10. p16: "Interestingly, the FRET ratio varied between cells, even under non-stress conditions (Fig. 4a,b), and correlated with sensor expression (Supplementary Fig. 5a). These data suggested that sensor expression, and overall protein concentration, may correlate with macromolecular crowding in the cell. In support of this hypothesis, experiments using purified SED1 found no correlation between SED1 concentration and FRET ratio in vitro (Supplementary Fig. 5b)." - Does increase sensor expression really increase the total protein concentration significantly? An alternative explanation (and more plausible in my view) is that this is due to a higher fluorescent relative to the background scattering (which is mostly in the donor channel). One way to tell is to check whether donor life time correlates with total fluorescence intensity in the life-time imaging data.

Reviewer #2:

Remarks to the Author:

A sensor to sense osmotic changes was constructed using an Arabidopsis LEA inserted between mCerulean3 as donor and mCitrine as the acceptor = FRET system. The sensor seems to work nicely in yeast, E. coli, humans' cells and tobacco, for sensing osmotic stressor such as PEG, NaCl, or water deficit. Yeast Mutants in hog (high osmolarity glycerol) pathway and pbs2, which is a scaffold MAPKK that integrates the two branches of the HOG pathway were used and sensor response were as expected.

However, surprisingly sensor did not work in Arabidopsis, authors explain that because the interaction with endogenous binding patterns and/or posttranslational modifications. LEA could be hyper-phosphorylated, but: LEA sequence was analysed for phosphorylation sites? If were because phosphorylation effect, why it worked in tobacco? phosphorylation from one plant to another is different? Yeast and human cells do not carry out the phosphorylation processes?

In conclusion, authors have a nice sensor for osmotic stress in cells, based on the ability of LEA proteins to change conformation due to osmotic pressure in cell. The sensor will be useful to test different stressors in cells and create macromolecular crowding maps. It was useful to test the known mechanisms such as HOG and pbs2, but, how novel mechanisms could be discovery using this sensor? Lots of mutants should be analysed to test new responses pathways to a particular stressor.

SED1 will be a fancy tool to test stressors and shows nicely how IDR can change structure; but cell responses to the stressors still will be needed to use other tools to find out the molecular mechanisms of cell responses.

In general the present report shows a very nice work using the ability of IDRs proteins

Reviewer #3:

Remarks to the Author:

This reports the development and deployment of a biosensor that uses the sensitivity of conformational equilibria of a specific intrinsically disordered protein to respond to changes in osmolarity. The fluorescence based sensor uses FRET between tethered donors and acceptors, with the conformational equilibria of the tethers being responsive to changes in osmolarity. The authors

use the disordered protein based on the plant protein LEA, and they show, rather convincingly, that the change in osmolarity / crowding leads to a conformational transition that is registered as ratiometric FRET in vitro, in budding yeast, and in different cell types. The authors also show that there appears to be sequence specificity to the osmolarity dependent conformational response, and that this sequence specificity confounds the use of SED1 in cell lines from which the LEA protein was derived. The suggestion is that the sensor will need to be deployed in orthogonal settings and it could be a very useful way to measure physiological states in a variety of unicellular and multicellular systems. The findings are novel and comprehensive and it represents a significant advance that will likely become a valuable tool for many labs that are interested in the impact of changes to osmolarity within cells. Several fields are likely to benefit from the advances reported here. Of course, one can always be persnickety and find nits to pick out, but that would defeat the purpose of peer review. The work, more or less as it stands, should be an invaluable addition to the literature. There are a few semantic issues and conceptual points that could use some clarification, but these are not absolutely essential. Therefore, since there is a first for everything, I am happy to report that I have no major issues to bring to the attention of the authors. This is a timely contribution that deserves to be published with some attention paid to cleaning up some semantic confusions.

Reviewer #4:

Remarks to the Author:

The study entitled "Intrinsically disordered protein biosensor tracks the physical-chemical effects of osmotic stress on cells" by Cuevas-Velazquez et al. introduces a novel biosensor to investigate rapid changes of intracellular environments. The sensor is constructed from a naturally occurring LEA-IDP and characterized in vitro e.g. in comparison to scrambled sequences and another crowding sensor. Its applicability is demonstrated in several types of cells, leading to novel biological insight and knowledge such as the size-effect of vacuoles as a water resort on the individual cell level. The ease of use of the sensor (by transfection and imaging) will attract a broad audience of scientists studying biopolymers in vivo. As such the paper is very well written and amenable to a broad audience. I recommend publication in NatCommun but I have one major point of criticism that the authors should address before publication:

The authors show how they calibrate their sensor in vitro, using some conditions like NaCl or PEG. However the sensor is responsive to both of these conditions, as expected, but I miss a more detailed characterization in vitro to show what changes in the cellular environments the sensor actually detects? What other salts were tested? Molecular vs macromolecular crowders? pH? Metabolites?

Based on this I also miss a discussion that compares the sensor to previous approaches, beyond the one by Boersma. Actually the same lab also developed an ionic strength sensor (FRET Analysis of Ionic Strength Sensors in the Hofmeister Series of Salt Solutions Using Fluorescence Lifetime Measurements). Other labs have also developed and applied osmotic sensors (e.g. Imperfect crowding adaptation of mammalian cells towards osmotic stress and its modulation by osmolytes). In particular the Schuler group recently published a paper of a single-molecule study of the IDP ProTa in the cytosol of HeLa cells under different osmotic conditions. Addressing this point is crucial if the sensor is applied under different cellular conditions in health and disease and will further improve the manuscript.

REVIEWER COMMENTS

Reviewer #1:

Comment 1.1

Cuevas-Velasquez et al. describe a new fluorescent biosensor based on the environmental sensitivity of intrinsically disordered regions, and show that it can be used to monitor osmotic stress in bacteria, yeast and some plants and cultured human cells. The sensor shows a significant improvement relative to existing crowding sensors, and represents a new type of biosensor design that will likely lead to many follow-up studies. The manuscript presents a multifaceted characterization of the sensor using both *in silico*, *in vitro* and *in vivo* experiments and represents both a significant advance in fundamental understanding (vacuole-size correlates with stress resistance) and provides a useful new research tool. There are many avenues that could be pursued further, but in most cases these are better left for follow-up studies.

Comment 1.2

There are, however, a few topics that are so central to this paper that I think it should be addressed before publication: It is unclear exactly what the biosensor measures, what the dynamic range is, and whether the reported FRET values can be compared between different experiments. These questions could either be addressed through additional experiments, analysis or discussion depending on the authors' judgement.

Response 1.2

We acknowledge the reviewer's comments and concerns and we provide a point by point response below.

Comment 1.3

Major points:

1. What does the sensor respond to? As the authors note, osmotic stress introduces several changes to the chemical environment at once, and it is important to clarify which are responsible for the compaction. In Fig. 2B it is convincingly shown that crowding is part of the answer. Another factor that is likely to have an impact is ionic strength as the compaction of IDPs are highly sensitive to electrostatic interactions. It would be nice to know how the purified reporter responds to ionic strength and possibly different solutes (using a similar setup to Ref. 19). Also, does the crowding effect observed in Fig. 2B explain the entire change seen in the osmotic response in cell experiment or is something missing? (It is difficult to compare the values in Fig. 2B to the remaining FRET reporter values as they are reported in different units).

Response 1.3

As the reviewer correctly points out, there are various effects that contribute to SED1 response in cells. In the first submission, we have demonstrated macromolecular crowding as one of the main contributions. In response to the reviewer's comments, we have conducted a solution space scan (similar to ¹) of both SED1 and CS (Response Fig. 1). This data is now replacing the original Fig. 2b, which we moved to Supplementary Fig. 4b. As can be seen, AtLEA4-5 clearly displays prominent change in FRET signal in response to crowding agents such as PEG and Ficoll. In addition, certain amine-derivative osmolytes such as TMAO, glycine, and sarcosine (but notably not betaine) induce a strong compaction effect. In terms of ionic strength, significant changes are only observed at concentrations > 1 M for both KCl and NaCl. Even then, the effect is smaller than the effect of large macromolecular crowders, which is significantly stronger at mM concentrations for PEG or Ficoll.

While this comparison showcases the dynamic range and sensitivity of the SED1 biosensor *in vitro*, it is difficult to extrapolate these scans to the actual cellular environment. Previous papers have done so by comparing an effective "volume fraction" of polymeric crowders. However, such comparisons can be highly misleading and an oversimplification of the intracellular effects, especially with respect to measurements of intrinsically disordered sequences. With the new results, we can conclude that SED1 FRET signal is dependent on macromolecular crowding and high osmolarity, but not to changes in ionic strength that may occur in most non-extremophile cell types. The interpretation of these results has been included in lines 219-225 of the revised manuscript.

Response Fig. 1. Experimental solution space scan of AtLEA4-5 and CS. Open circles show the normalized FRET ratio (DxA_m/DxD_m) for the indicated concentration of each solute, with two points (that often overlap) for each concentration taken from separate repeats, highlighting the reproducibility of the data. Background color intensity represents sensitivity of change to solute addition. Stronger colors indicate stronger sensitivity. Red: compaction;

blue: expansion; white: no change. Solution concentrations are given in weight percent (0-25 or 0-12 wt %) or molar (0-1.5 M). This figure is now Fig. 2b of the revised manuscript.

Comment 1.4

2. Can the sensor be saturated? Fig. 1B shows a plateau in the FRET ratio at high osmolyte values. Does this represent an intrinsic plateau in the dynamic range of the sensor or a plateauing in the osmotic response of the cells. One way to answer this would be to clarify whether there are signs of saturation in the experiments performed with the purified sensor.

Response 1.4

We thank the reviewer for this important question. As described in detail in “*Response 1.3*”, we performed *in vitro* solution space scanning analysis based on the reviewer’s suggestion. The data shows that the plateau observed in yeast cells likely corresponds to the osmotic response of the cells. While we fully expect SED1 to have a point at which signal is saturated (where the end-to-end distance is sufficiently longer or shorter than R_0 for the FRET pair), the *in vitro* data shows a different behavior. In the range of concentrations we have tested *in vitro*, there are few notable solutes that do not display a linear dependence of FRET ratio on solute concentration. None of these appear to have been saturated (see Response Fig. 2 for PEG 6,000). Specifically, non-monotonic trends are observed for certain charged species, including NaCl, KCl, and at higher concentrations sarcosine. This is in line with the well-documented effect of charged solutes, where at low concentrations the effect is dominated by electrostatic screening, and at high concentrations the effect is dominated by specific ion effects². This effect has also been shown to play a dominant role for IDRs, specifically in two recent works^{1,3}. For macromolecular crowders, there appears to be an inflection point resulting in a much higher FRET ratio at higher concentration of PEGs. Closer inspection revealed that at this point SED1 crashes out of solution, resulting in an uninterpretable FRET signal (we added a sentence describing this in the Supplementary Fig. 4 legend). This behavior has been seen previously for similar FRET constructs in high concentrations of PEG¹. We can therefore only state that this saturation point is not reached in any of the *in vitro* conditions we have tested. Notably, this behavior has happened for the CS sensor as well in sufficiently high PEG concentrations. In line with our “*Response 1.3*”, this aggregation is not observed in any cell type as far as we can discern with our imaging setups, further validating our statement that PEG concentrations cannot be realistically compared to the intracellular environment.

Response Fig. 2. Fluorescence emission spectra of purified recombinant SED1 in the presence of increasing concentrations (wt %) of PEG 6,000 (shades of blue). Fluorescence emission spectra of SED1-expressing yeast cells treated with 1.4M NaCl is included as a reference (red). Fluorescence values were normalized to the value at 515 nm. The dramatic drop in donor signal at 20 and 24 wt % PEG 6,000 are the point at which SED1 begins to aggregate.

Comment 1.5

3. Analysis of FRET data. Through-out the manuscript the fluorescent read-out as “normalized $DxAm/DxDm$ ”. As I understand the methods description, these are the raw fluorescence values and are not corrected for back-scattering, direct excitation of the acceptor and bleed-through of the donor emission into the acceptor signal (which will be significant for these FPs). If I understand this correctly, then this means that a change from “normalized $DxAm/DxDm$ ” 1 to 1.3 will represent something different for different experiments, and mean that the read-out from the biosensor should not be directly compared except for very similar experiments. I think the authors should consider whether it would be possible to correct the fluorescence signals to get values that are more comparable across various setups. Also, the normalization may be needed to get clean comparisons across a range of experiments, but removes potentially useful information. I think you should consider either presenting unnormalized data, or alternatively present the base line FRET ratio for each normalized value to allow the reader to judge which normalized values are comparable.

Response 1.5

We agree with the reviewer’s point that normalization is necessary for comparing SED1 read out for a particular set of experiments, but does not allow the comparison between different cell types. We think that readout normalization is important to show the magnitude of the change imposed by the osmotic shock, which is what we aim to demonstrate in this work. Cells from different organisms have substantially different intracellular physical-chemical properties, so showing the unnormalized FRET ratio might lead to mistaken interpretations. We decided to maintain the normalized FRET ratios but, given the importance of showing the unnormalized values for the different experiments throughout the manuscript, we have included

Supplementary Fig. 11 with the unnormalized FRET values of untreated cells (Response Fig. 3). This data confirms that the standard/isosmotic base line FRET levels are different for *E. coli*, *S. cerevisiae*, *N. benthamiana*, and *H. sapiens* cells. With these results, the reader can judge how comparable are the normalized values. The reference to this data has been included in lines 408-409 of the revised manuscript.

Response Fig. 3. FRET ratio (DxAm/DxDm) of SED1 in the indicated context. Purified recombinant full-length SED1 in buffer (20 mM sodium phosphate buffer, 100 mM NaCl, pH 7.4). Standard/isosmotic conditions FRET ratios are shown for the different organisms.

Comment 1.6

4. It is a little concerning and inconvenient that the sensor partition to stress granules in yeast (Fig. S5c). The 10% 1,6-hexanediol control shows that the increase in FRET is not due to this partitioning, however the FRET ratio is significantly higher in the presence of 1,1-hd (Fig. S5d). It would be beneficial if this was discussed more fully, as this potentially raises the question of what environment the sensor samples. Also, it would be useful if the authors could state whether something similar happens in all the cell types they have tested, or whether it is specific to yeast.

Response 1.6

We agree with the reviewer's concern that SED1 recruitment to liquid-like granules in yeast complicates the interpretation of the observed changes in FRET under hyperosmotic stress. We do believe that we can confidently say, however, that 1,6-hexanediol treatment demonstrates that the increase in FRET is not caused by protein condensation. The more difficult result to interpret is the observed increase in the FRET ratio of hyperosmotically treated cells in the

presence of 1,6-hexanediol. This result could be caused by a number of phenomena which are not within the scope of this work. 1,6-hexanediol treatment has been shown to induce several nonspecific cellular processes⁴. 1,6-hexanediol also dissociates endogenous biomolecular condensates, which would likely alter the physiological state of cells. We think that at this point it would be highly speculative to discuss this result other than demonstrating that the FRET change in cells is not the result of SED1 partitioning into granules. Moreover, this phenomenon is only observed in yeast cells. We have included a passage in the results section to state that SED1 recruitment to granules is only observed in yeast cells (line 299 of the revised manuscript).

Comment 1.7

5. Clear discussion of limitations. The discussion currently focus on the potential applications of the sensor, and largely omits discussion of the limitations of the sensor. I think your readers will benefit from paragraph dedicated to the limitations of the sensor, where you clearly lay out what can and cannot be concluded (drawing on the points above). I would like to point out that I think that this paper clearly merits publication in a good journal, and that an explicit recognition of limitation of the study does not detract from this.

Response 1.7

We thank the reviewer for this suggestion. We have included the following lines in the discussion about the biosensor limitations.

“The main caveat of SED1 is the lack of responsiveness in the model plant Arabidopsis thaliana.”

This line complements the already discussed limitation of SED1 unresponsiveness in Arabidopsis, but it makes the limitation now explicit. This has been included in lines 441-442 of the revised manuscript.

“The first generation of genetically-encoded biosensors often come with a handful of limitations. Subsequent biosensor optimization focuses on larger dynamic range, improved specificity, higher brightness, and fewer undesired side-effects. It is very likely that future rounds of SED1 improvement would be designed specifically for each kind of organism where it is to be used. The use of computational solution space scanning will likely aid in these efforts as the role of individual amino acid variants can be tested”. This paragraph emphasizes the expected limitations of a first-generation biosensor observed in SED1. We also propose some strategies for future optimization rounds. This has been included in lines 434-439 of the revised manuscript.

Comment 1.8

6. The authors use the term “plethora” in both the introduction and discussion. In my minds, this term has negative connotations and thus likely do not reflect what the authors mean, and you may wish to consider a different wording.

Response 1.8

We have changed the phrase “*a plethora of*” to “*various*” in the indicated sections (lines 62 and 412 of the revised manuscript).

Comment 1.9

7. p. 6 “The FRET ratio change was significantly smaller when we tested a globular protein (arabinose-binding protein, ABP) as a reference” - I do not understand this control. Maybe you want to elaborate on why this protein was chosen.

Response 1.9

The protein conformation of IDRs has been shown to be more sensitive to environmental perturbations than their globular counterparts^{1,5}. We chose ABP because it is a globular protein that has been successfully used in a biosensor of a small molecule (arabinose) with a small K_d ⁶. Based on the two aforementioned properties, we expected that ABP would be a globular protein insensitive to high osmolarity and/or macromolecular crowding in cells, which is demonstrated in the manuscript. We included a description of this in lines 125-128 of the revised manuscript.

Comment 1.10

8. The FRET efficiencies from FLIM should probably not be reported with 4 significant digits.

Response 1.10

We have changed the FRET efficiency values to 2 significant digits with standard deviation (lines 313 and 336 of the revised manuscript).

Comment 1.11

9. How were the scrambled sequences exactly constructed? By manually or using an algorithm? Were other parameters also considered e.g. the degree of charge mixing? (Can be quantified from the CIDER web server).

Response 1.11

The Scrambled sequences were randomly-designed using the Scrambler tool of PeptideNexus (<https://peptidenexus.com/>). We specifically looked for sequences that keep the high disorder prediction but with a decreased helical propensity, as indicated in the results section of the first submission. We also selected scrambled versions with a smaller degree of charge mixing relative to AtLEA4-5 (larger Kappa value) (Response Fig. 4). Based on the reviewer’s suggestion, we have indicated this in lines 168-170 of the revised manuscript and we included the CIDER parameters in Supplementary Fig. 3c.

Protein	Length (aa)	Kappa	FCR	Omega (polar)	Omega (hydrophobic)
AtLEA4-5	158	0.061	0.203	0.214	0.186
Scramble-1	158	0.174	0.203	0.216	0.063
Scramble-2	158	0.164	0.203	0.225	0.099
Scramble-3	158	0.177	0.203	0.321	0.096
Scramble-4	158	0.186	0.203	0.272	0.104
Scramble-5	158	0.21	0.203	0.256	0.11

Response Fig. 4. CIDER parameters of AtLEA4-5 and the five different scrambled versions. Kappa: Measure of the extent of charged residues (R,K,E,D) segregation; FCR: Fraction of charged residues; Omega (polar): Measure of the extent of polar residues (Q,N,S,T,G,H,C) segregation; Omega (hydrophobic): Measure of the extent of polar residues (A,L,M,I,V) segregation.

Comment 1.12

10. p16: “Interestingly, the FRET ratio varied between cells, even under non-stress conditions (Fig. 4a,b), and correlated with sensor expression (Supplementary Fig. 5a). These data suggested that sensor expression, and overall protein concentration, may correlate with macromolecular crowding in the cell. In support of this hypothesis, experiments using purified SED1 found no correlation between SED1 concentration and FRET ratio in vitro (Supplementary Fig. 5b).” – Does increase sensor expression really increase the total protein concentration significantly? An alternative explanation (and more plausible in my view) is that this is due to a higher fluorescent relative to the background scattering (which is mostly in the donor channel). On way to tell is to check whether donor life time correlates with total fluorescence intensity in the life-time imaging data.

Response 1.12

Based on the reviewer’s suggestion, we performed the lifetime versus fluorescence intensity correlation analysis. We found a significant negative correlation between these parameters, in agreement with our hypothesis (Response Fig. 5). We have now included the lifetime-fluorescence intensity correlation in Supplementary Fig. 5b.

Response Fig. 5. Pearson's correlation of fluorescence lifetime of the donor and mean acceptor fluorescence intensity (kCounts) of individual live yeast cells. $r = 0.671$, $p\text{-value} = 2 \times 10^{-14}$. This figure is now Supplementary Fig. 5b.

Reviewer #2:

Comment 2.1

A sensor to sense osmotic changes was constructed using an Arabidopsis LEA inserted between mCerulean3 as donor and mCitrine as the acceptor)= FRET system. The sensor seems to work nicely in yeast, E. coli, humans' cells and tobacco, for sensing osmotic stressor such as PEG, NaCl, or water deficit. Yeast Mutants in hog (high osmolarity glycerol) pathway and pbs2, which is a scaffold MAPKK that integrates the two branches of the HOG pathway were used and sensor response were as expected. However, surprisingly sensor did not work in Arabidopsis, authors explain that because the interaction with endogenous binding patterns and/or posttranslational modifications. LEA could be hyper-phosphorylated, but: LEA sequence was analysed for phosphorylation sites? If were because phosphorylation effect, why it worked in tobacco? phosphorylation from one plant to another is different? Yeast and human cells do not carry out the phosphorylation processes?

Response 2.1

We appreciate the reviewer's interest in implementing SED1 in Arabidopsis, as this was also an area where we had placed a significant amount of effort. AtLEA4-5 has been shown to be hyper-phosphorylated in Arabidopsis in at least 16 of its 33 Ser, Thr and Tyr residues⁷. As an intrinsically disordered protein, AtLEA4-5 is prone to be heavily post-translationally-modified, which might also happen in cells of other organisms, as pointed out by the reviewer. Indeed, SED1 unresponsiveness in Arabidopsis could be caused by a number of reasons including interaction of AtLEA4-5 with its endogenous binding partners, as stated in the discussion section of the first submission. While we have not directly examined *in vivo* phosphorylation states of the protein

across different contexts, it may be that in the tobacco transient assay phosphorylation of the protein occurs less. As a follow up study, we are currently engineering SED1 to make it responsive in Arabidopsis which is a primary goal for our research, but not the subject of the present manuscript.

Comment 2.2

In conclusion, authors have a nice sensor for osmotic stress in cells, based on the ability of LEA proteins to change conformation due to osmotic pressure in cell. The sensor will be useful to test different stressors in cells and create macromolecular crowding maps. It was useful to test the known mechanisms such as HOG and psb2, but, how novel mechanisms could be discovery using this sensor? Lots of mutants should be analysed to test new responses pathways to a particular stressor. SED1 will be a fancy tool to test stressors and shows nicely how IDR can change structure; but cell responses to the stressors still will be needed to use other tools to find out the molecular mechanisms of cell responses.

In general the present report shows a very nice work using the ability of IDRs proteins

Response 2.2

We agree that follow up studies will likely make good use of this tool to investigate new mechanisms of osmotic stress perception and response. We also agree that classical tools for studying stress responses will be required to get the full picture of the sensing mechanisms, but the availability of a biosensor that dynamically reports the physical-chemical effects of osmotic stress with subcellular resolution, in real time, and in a non-destructive manner will be a game-changer in the field.

Reviewer #3:

Comment 3.1

This reports the development and deployment of a biosensor that uses the sensitivity of conformational equilibria of a specific intrinsically disordered protein to respond to changes in osmolarity. The fluorescence based sensor uses FRET between tethered donors and acceptors, with the conformational equilibria of the tethers being responsive to changes in osmolarity. The authors use the disordered protein based on the plant protein LEA, and they show, rather convincingly, that the change in osmolarity / crowding leads to a conformational transition that is registered as ratiometric FRET in vitro, in budding yeast, and in different cell types. The authors also show that there appears to be sequence specificity to the osmolarity dependent conformational response, and that this sequence specificity confounds the use of SED1 in cell lines from which the LEA protein was derived. The suggestion is that the sensor will need to be deployed in orthogonal settings and it could be a very useful way to measure physiological states

in a variety of unicellular and multicellular systems. The findings are novel and comprehensive and it represents a significant advance that will likely become a valuable tool for many labs that are interested in the impact of changes to osmolarity within cells. Several fields are likely to benefit from the advances reported here. Of course, one can always be persnickety and find nits to pick out, but that would defeat the purpose of peer review. The work, more or less as it stands, should be an invaluable addition to the literature. There are a few semantic issues and conceptual points that could use some clarification, but these are not absolutely essential. Therefore, since there is a first for everything, I am happy to report that I have no major issues to bring to the attention of the authors. This is a timely contribution that deserves to be published with some attention paid to cleaning up some semantic confusions.

Response 3.1

We acknowledge the reviewer's encouraging comments. We have screened the manuscript for semantic confusions and made edits when we saw fit.

Reviewer #4:

Comment 4.1

The study entitled "Intrinsically disordered protein biosensor tracks the physical-chemical effects of osmotic stress on cells" by Cuevas-Velazquez et al. introduces a novel biosensor to investigate rapid changes of intracellular environments. The sensor is constructed from a naturally occurring LEA-IDP and characterized in vitro e.g. in comparison to scrambled sequences and another crowding sensor. Its applicability is demonstrated in several types of cells, leading to novel biological insight and knowledge such as the size-effect of vacuoles as a water resort on the individual cell level. The ease of use of the sensor (by transfection and imaging) will attract a broad audience of scientists studying biopolymers in vivo. As such the paper is very well written and amenable to a broad audience. I recommend publication in NatCommun but I have one major point of criticism that the authors should address before publication:

The authors show how they calibrate their sensor in vitro, using some conditions like NaCl or PEG. However the sensor is responsive to both of these conditions, as expected, but I miss a more detailed characterization in vitro to show what changes in the cellular environments the sensor actually detects? What other salts were tested? Molecular vs macromolecular crowders? pH? Metabolites?

Response 4.1

Based on the reviewer's suggestion, we have conducted a solution space scan (similar to ¹) of both SED1 and CS (Response Fig. 1). This data is now replacing the original Fig. 2b, which we moved to Supplementary Fig. 4b. As can be seen, AtLEA4-5 clearly displays prominent change in

FRET signal in response to crowding agents such as PEG and Ficoll. In addition, certain amine-derivative osmolytes such as TMAO, glycine, and sarcosine (but notably not betaine) induce a strong compaction effect. In terms of ionic strength, significant changes are only observed at concentrations > 1 M for both KCl and NaCl. Even then, the effect is relatively small compared to the effect of large macromolecular crowders, which is significantly stronger even at mM concentrations for PEG or Ficoll. The interpretation of these results has been included in lines 219-225 of the revised manuscript.

Response Fig. 1. Experimental solution space scan of AtLEA4-5 and CS. Open circles show the normalized FRET ratio (Dx_{Am}/Dx_{Dm}) for the indicated concentration of each solute, with two points (that often overlap) for each concentration taken from separate repeats, highlighting the reproducibility of the data. Background color intensity represents sensitivity of change to solute addition. Stronger colors indicate stronger sensitivity. Red: compaction; blue: expansion; white: no change. Solution concentrations are given in weight percent (0-25 or 0-12 wt %) or molar (0-1.5 M). This figure is now Fig. 2b of the revised manuscript.

Comment 4.2

Based on this I also miss a discussion that compares the sensor to previous approaches, beyond the one by Boersma. Actually the same lab also devolved an ionic strength sensor (FRET Analysis of Ionic Strength Sensors in the Hofmeister Series of Salt Solutions Using Fluorescence Lifetime Measurements). Other labs have also developed and applied osmotic sensors (e.g. Imperfect crowding adaptation of mammalian cells towards osmotic stress and its modulation by osmolytes). In particular the Schuler group recently published a paper of a single-molecule study of the IDP ProT α in the cytosol of HeLa cells under different osmotic conditions.

Addressing this point is crucial if the sensor is applied under different cellular conditions in health an disease and will further improve the manuscript.

SE

Response 4.2

We thank the reviewer for these references, and have included them in our introduction and discussion. Overall, in comparison to Schuler⁸, we see similar effects for SED1 and ProTα *in vitro* - compaction is notable only for larger polymeric crowders. However, expanding to more types of solutes reveal more intricacies in this behavior as smaller molecules such as TMAO and sarcosine also induce a notable compaction effect. In-cell, a similar effect is also observed - as osmotic pressure is exerted, the chain expands, and for SED1 we need not use extensive osmotic pressures - for the addition of ~ 300 mOsm the volume change is roughly 20% and the effect, at least in mammalian cells, is noticeable, as shown in Fig. 5d,e. In Schuler's work, the authors reported a noticeable effect after a loss of 50% of cellular volume (but only this condition is reported). This bodes well for the sensing properties of SED1.

Comparison with Ebbinghaus' work⁹ also shows similar trends *in vitro*, though the behavior *in vivo* does match up in some aspects (larger crowders induce a stronger effect), it does not in others - TMAO for example induced a relatively strong compaction effect for SED1. In addition, the intracellular environment of most organisms displayed a marked change in FRET signal compared to the recombinantly expressed protein in buffer. We attribute this to the heteropolymeric nature and solute-specific interactions of SED1 compared to the homopolymeric nature of PEG.

While comparison to other FRET biosensors is valuable, the focus of this paper is not on comparison with other biosensors but rather the novel use of IDRs as sensors of the intracellular environment. In addition, there are some fundamental differences between the biosensor reported here (and Boersma's CS) to the two mentioned by the reviewer: Both Ebbinghaus polymer-based sensor and Schuler's FRET-labeled IDP rely on *in vitro* labeling and subsequent microinjection of the labeled construct into the cell. While this presents an advantage in terms of the fluorescent properties of the construct and good control over intracellular concentration of the construct, this process requires expertise in both labeling and microinjection, creating a barrier for other labs interested in using the biosensors. In addition, the reliance on injection drastically diminishes the throughput of this approach. We have included a discussion on this regard in lines 453-462.

RESPONSE REFERENCES.

1. Moses, D. *et al.* Revealing the Hidden Sensitivity of Intrinsically Disordered Proteins to their Chemical Environment. *J. Phys. Chem. Lett.* **11**, 10131–10136 (2020).
2. Pegram, L. M. *et al.* Why Hofmeister effects of many salts favor protein folding but not

- DNA helix formation. *Proc. Natl. Acad. Sci. U. S. A.* **107**, 7716–7721 (2010).
3. Vancaenenbroeck, R., Harel, Y. S., Zheng, W. & Hofmann, H. Polymer effects modulate binding affinities in disordered proteins. *Proc. Natl. Acad. Sci. U. S. A.* **116**, 19506–19512 (2019).
 4. Düster, R., Kaltheuner, I. H., Schmitz, M. & Geyer, M. 1,6-Hexanediol, commonly used to dissolve liquid-liquid phase separated condensates, directly impairs kinase and phosphatase activities. *J. Biol. Chem.* **296**, 100260 (2021).
 5. Holehouse, A. S. & Sukenik, S. Controlling Structural Bias in Intrinsically Disordered Proteins Using Solution Space Scanning. *J. Chem. Theory Comput.* **16**, 1794–1805 (2020).
 6. Kaper, T., Lager, I., Looger, L. L., Chermak, D. & Frommer, W. B. Fluorescence resonance energy transfer sensors for quantitative monitoring of pentose and disaccharide accumulation in bacteria. *Biotechnol. Biofuels* **1**, 11 (2008).
 7. Mergner, J. *et al.* Mass-spectrometry-based draft of the Arabidopsis proteome. *Nature* **579**, 409–414 (2020).
 8. König, I., Soranno, A., Nettels, D. & Schuler, B. Impact of in-cell and in-vitro crowding on the conformations and dynamics of an intrinsically disordered protein. *Angew. Chem. Weinheim Bergstr. Ger.* **133**, 10819–10824 (2021).
 9. Gnutt, D., Brylski, O., Edengeiser, E., Havenith, M. & Ebbinghaus, S. Imperfect crowding adaptation of mammalian cells towards osmotic stress and its modulation by osmolytes. *Mol. Biosyst.* **13**, 2218–2221 (2017).

Reviewers' Comments:

Reviewer #1:

Remarks to the Author:

The authors have addressed all my concerns and I recommend publication without further delay.

Reviewer #2:

Remarks to the Author:

Authors have answered very clear all comments

I do not have further questions or comments

Reviewer #4:

Remarks to the Author:

The authors addressed my concerns by further experiments and discussion and I recommend publication.